# Depth Extrapolation of Decoders Trained on Nested Structures

## Abstract

Reasoning problems with deeply nested formal statements are challenging for humans and machines alike. We investigate how next-token predictors learn such structures, and whether they extrapolate to more deeply nested cases, within a single inference pass. A case study of Boolean logic simplification demonstrates that a specialized decoder Transformer seems to perform well when it overfits, but fails at extrapolating. To understand if this limitation is universal, we propose a theoretical grounding of memorization in a self-attention head. We apply this theory to a simpler problem: completion of a bounded stack of parentheses. From the theoretical construction we derive a closed-form model that perfectly fits a single sequence training set. We prove that it also completes any out-of-sample parentheses prefix, regardless of the context depth. In contrast, we observe that decoder Transformers trained with gradient descent on this task fail at depth extrapolation. Gradient-trained decoders demand large samples and a high-dimensional embedding space to achieve high accuracy on test sets nearly as deep as the training set. However, when the gap between training and test depths widens, gradient-trained models fail.

## 1 Introduction

Decoder Transformers Radford et al. (2018) exhibit emerging reasoning capabilities (Wei et al., 2022a; Polu and Sutskever, 2020; Bubeck et al., 2023; Trinh et al., 2024; Reid et al., 2024). Extrapolating reasoning capabilities to tasks that are harder than already solved problems would unblock major scientific breakthroughs. Measuring whether a machine can reason on problems of unbounded difficulty raises fundamental questions about fair assessment of models' performance. Measurements of basic capabilities of neural networks has historically followed the methodology of evaluations on independently and identically distributed unseen data (Vapnik, 1998). However, the internet-extracted massive training sets cover most of human knowledge. Therefore, many reported evaluations of generalization possibly use non-independent test data, raising questions about the validity of the reported performances Mialon et al. (2023); Schaeffer et al. (2024). In a parallel stream of work, (Delétang et al., 2023; Kazemnejad et al., 2024) have studied Transformers' extrapolation to larger context windows. To study extrapolation to harder-than-observed reasoning problems, we focus on decoder Transformers' handling of nested structures (*e.g.* Boolean expressions), where we use depth as a measure of complexity. We ask: *do decoder Transformers trained on data nested up to depth $q$ perform well on deeper test data?*

In a motivating case study in Section 2, we experiment on Boolean expression simplification (McCluskey, 1956). The synthetic data we generate are minimal examples of reasoning with nested structures. We observe that general purpose Large Language Models (LLMs) struggle with such nested data as the test data gets deeper, especially if they use a single pass of generation. Our experiments on a controlled dataset suggest that memorization is a primary driver of performance: in-sample performance far exceeds out-of-sample's. But overfit models fail at depth extrapolation. These findings raise multiple questions: is memorization the only mechanism to handle nested structures? We know that generalization in the standard sense is not in conflict with memorization (Belkin, 2021; Stephenson et al., 2021; Tirumala et al., 2022; Feldman, 2020), but what about extrapolation: can a Transformer that fits the data perfectly also reliably extrapolate to more deeply nested problems? The theoretical results presented in Section 3 pinpoint a regime where models perfectly fit the training data and are also stable in unseen data.

We apply the proposed construction to a simple Dyck language. Dyck languages encapsulate the barebone of hierarchy in Boolean expressions and other nested structures, and are also used to represent context-free grammars (Chomsky, 1956). In Section 4 we build a solution of next-token prediction trained on *a single sequence*. The closed-form solution not only memorizes the training sequence, but also solves the balanced parentheses completion problem on more nested out-of-sample data. This suggests that Transformers with a single generation pass are not universally incapable of depth extrapolation. In contrast, our numerical experiments show that the *gradient-trained* Transformers fail at finding such a solution. Instead, the trained models demand lots of data, high embedding dimensions and many layers for fitting the training set and performing in-sample completion well. The model also performs well on out-of-sample test sets of lower or equal depth, yet it fails at generalizing to deeper samples. This indicates an implicit bias of gradient-based training to finding solutions with limited depth extrapolation.

## 2 Transformers simplify nested Boolean expressions when they overfit. Do they also extrapolate?

Simplifying entangled information into simpler logic is a critical capability for reasoning on complex tasks. Simple expressions reconcile conflicts in complex logic. They are easier to understand and verify by humans, and require lower effort for human or a machine to answer questions about the underlying data. To simplify Boolean expressions, McCluskey (1956) relies on the truth table of the expression, and performs searches in the exponentially large (in the variable count) stored table for prime implicants. The more practical 'simplifying inside-out' method suffers exponential computation in the expression's depth. State-of-the-art LLMs fail at this task when the expression's depth is large in a single inference step. If multiple inferences are allowed, they articulate a chain-of-thought which follows the inside-out logic, but small errors often propagate throughout the chain, leading to incorrect conclusions. Here we examine the outcome of a single inference pass of a (pre-)trained decoder Transformer on data of known and bounded depth. We compare in-sample versus out-of-sample performance, when data is stratified by depth. We emphasize that our goal is not to build the best inference logic to solve the problem efficiently. Our goal is to show-case depth extrapolation (or lack thereof) of a next-token predictor.

### 2.1 Data generation and stratification along depth

We consider Boolean expressions formed using a random generator from a fixed number of variables $p0, p1, p2, p3$ and operators $\&, |, \neg$ nested at a desired depth within parentheses. Our generator (pseudo-code in Appendix 5.1.2) creates Boolean expressions containing $k$ nested clauses of maximum depth $q$, by dynamically calling the `random_clause` generator of lower depth and joining `left` and `right` expressions of lower depth using either $\&$ or $|$ with equal probability. At depth zero, `random_clause` returns either one of the variables or its negation. We use `simplify_logic` from `sympy.logic.boolalg` as an oracle, to map each complex Boolean expression into a simplified expression. Each equality of a complex expression and its simplified version constitute one sequence in our dataset. For example, with $k = 3, q = 3$ one such sequence is

$$< \texttt{start} > (((( \, \texttt{p3} \, \& \, \texttt{p3}) \, \& \, (\texttt{true} \, | \, \neg\texttt{p2} \,)) \& \, (\neg\texttt{p2} \, \& \, \texttt{p0} \,))) \, | \, (((\neg\texttt{p3} \, \& \, \texttt{p1}) \, \& \, (\texttt{true} \& \, \neg\texttt{p2})))$$
$$\& \, ((( \, \texttt{p0} \, \& \, \texttt{p2}) \, \& \, ((\texttt{p3} \, | \, \texttt{true}) \, | \, (\neg\texttt{p3} \, | \, \neg \, \texttt{p3})))) \, = \, \texttt{p0} \, \& \, \texttt{p3} \, \& \, \neg \, \texttt{p2} < \texttt{end} >$$

The symbols $\{ \, \texttt{p0, p1, p2, p3}, < \texttt{start} >, < \texttt{end} >, \&, \, |, \, \neg, \, =, \, \texttt{false, true}, (, )\}$ are tokenized. We use a long concatenation of tokenized equalities to train models on.

### 2.2 Numerical findings on in-sample and out-of-sample deeper expressions

**Preliminary findings on logic simplification with LLMs.** As a point of reference, we prompt a general purpose LLMs to simplify a Boolean expressions (see Appendix 5.1.1). When a large stack of such expressions are passed, the model performs a single inference on each and gets almost all wrong. When the model is given a single simplification task, it often returns correct simplifications for shallow expressions of depth $q = 1$. However, as the expression gets deeper $q \geq 3$, even when articulating the correct chain-of-thoughts (Wei et al., 2022b), LLMs return mostly incorrect answers.
**Train and test split and model training.** We generate a random sample of Boolean expressions and

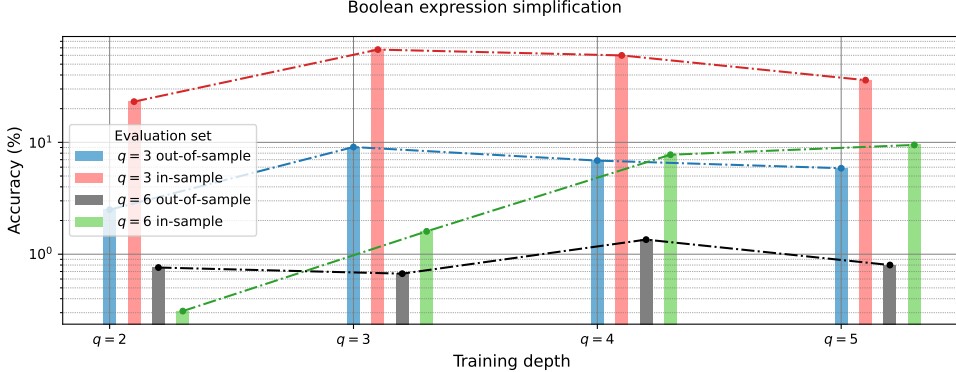

Figure 1: Accuracy of Boolean simplification (here in log-scale) shows the gap between in-sample and out-of-sample evaluations and also the failure to extrapolate to deeper test sets, here $q = 6$.

their simplified forms given 5 clauses and a maximum nested depth $q$. We define as *out-of-sample* all expressions with simplified form (right-hand-side of equality) in a held-out set of 128 randomly selected expressions. *In-sample* data are expressions with simplified forms excluded from the set of 128 held-out. We train a $d = 1024$ dimensional model with a context length of $n = 1024$, $h = 8$ heads and $L = 8$ layers: $\approx \cdot 10^8$ parameters in total. We train models for $10^5$ steps on $2^{14} \approx 1.6 \ 10^4$ batches from $10^9$ length sequences. We align the minibatches to start at $< \mathtt{start} >$ and be 1024 tokens long. The minibatches do not necessarily end on a $< \mathtt{end} >$ token. The exact values of these parameters are not chosen for performance optimization, as our goal in this section is simply to show-case tensions between memorization and depth extrapolation.

**Evaluation data and metrics.** To evaluate a model, we feed the model with expressions starting with $< \mathtt{start} >$ and ending with $=$. We generate next tokens until $< \mathtt{end} >$ is met. Provided that the number of tokens in the new expression is lower than in the tokenized left-hand-side, we decode the obtained expression and the true value. We use $\mathtt{equals}$ method from $\mathtt{sympy}$ to verify equality on $10,000$ samples. We report in Figure 1 accuracy of Boolean expression simplification for in and out-of sample expressions of depth $q = 3$ and $q = 6$.

**Observations.** We observe that the model trained on $q = 3$ data performs best at $q = 3$, for both out-of-sample (blue, $9\%$) and in-sample (red, $68\%$) testing. The in-sample evaluation task is designed to measure fit to data, or memorization and regurgitation of training sequences. We also note that accuracy reported for test sets of depth $q = 3$, both in and out-of sample, is the highest across different training depths when $q = 3$. This indicates that whether tested in-sample or out-of-sample, performance is best near the training set's depth. Finally, tests on deeper $q = 6$ examples show overall lower performance as compared to $q = 3$. The in-sample performance seems to increase as $q$ approaches the target value $q = 6$, while out-of-sample accuracy fluctuates around $1\%$.

We studied simplification of Boolean expressions an example of reasoning task. We observed that when successful, next-token predictors rely heavily on memorization. Overfit models have high in-sample performance but they fail at extrapolation to harder tasks with deeper nested expressions. Is this finding universal across all nested data? Is it a property of Transformer model architecture? Or is it an implicit bias of the training method we used? To answer these questions, in the next Section we review mechanisms of sequence fitting and generation in a single attention head. Section 4 applies the construction of Section 3 to the simplest example of nested structure: completion of Dyck words. For this problem (1) We demonstrate that a closed-form Transformer is capable of completing any bounded parenthesis sequence. (2) We empirically compare performance in terms of model size and sample complexity of the closed-form solution with gradient trained models, on deeper than in-sample test sets.

## 3 SEQUENCE FITTING AND GENERATION IN SELF-ATTENTION

In this section, we present a general theoretical framework to show the contribution of the added positional embeddings to sequence fitting and generation. We use a single-layer, single-head decoder

Transformer model Chorowski et al. (2014; 2015); Radford et al. (2019), equipped with added trained positional embeddings and tied unembedding head Edelman et al. (2022); Fu et al. (2023). This framing isolates the key functionality of converting variable-length sequences into fixed-dimensional vectors. We write a closed-form solution for the autoregressive loss with a single training point. The theory presented here demonstrates how learned positional embeddings fit to data, autoregress and generate next tokens. We reveal connections between the state variables: embedding dimension, context length, vocabulary length. In particular, the established relationship grounds memorization in large embedding spaces and indicates a stable regime for sequence generation. As an application of this theory, the expression of the optimal positional embedding gives a closed-form solution that we leverage in Section 4 to generate valid Dyck sequences, from any test prefix depths. We will see in our numerical experiments that while this depth extrapolation capable function lies in the solutions space, gradient trained models fail to find it.

## 3.1 NOTATIONS AND PROBLEM SETTING

We consider a vocabulary of $t$ tokens, represented by integers $[t] = \{1, \ldots, t\}$. We employ a decoder model to predict the next token given the sequence of previous $r \leq n$ tokens seen as as context, denoted by the tuple $\mathbf{s} = (\mathbf{s}_1, \ldots, \mathbf{s}_r)$. Each token is embedded as a $d$-dimensional vector, with $\mathbf{E} \in \mathbb{R}^{t \times d}$ stacking all embeddings as row vectors. The $i$-th row of this matrix is the embedding of token $i \in [t]$. We use $\mathbf{E_s} \in \mathbb{R}^{r \times d}$ to represent the embeddings of all tokens in context $\mathbf{s}$. On row $i \leq r$, the matrix $\mathbf{E_s}$ contains the embedding vector of token $\mathbf{s}_i$: $(\mathbf{E_s})_{i,:} = \mathbf{E}_{\mathbf{s}_i,:}$. We define the (row-wise) *softmax* operator of a matrix as $\mathtt{softmax}(\mathbf{X})_{i,j} = e^{\mathbf{X}_{i,j}} / \sum_k e^{\mathbf{X}_{i,k}}$, where we use the convention $\exp(-\infty) = 0$. We introduce the causal masking matrix $\mathbf{M}$, where $\mathbf{M}_{i,j} = 0$ if $i \leq j$ and $\mathbf{M}_{i,j} = -\infty$ if $i > j$. Given query/key/value matrices $\mathbf{Q}, \mathbf{K}, \mathbf{V} \in \mathbb{R}^{d \times d}$, the masked self-attention of $\mathbf{X} \in \mathbb{R}^{r \times d}$ is: $\mathtt{att}(\mathbf{X}; \mathbf{M}, \mathbf{Q}, \mathbf{K}) = \mathtt{softmax}(\mathbf{X}\mathbf{Q}\mathbf{K}^\mathsf{T}\mathbf{X}^\mathsf{T} + \mathbf{M}) \in \mathbb{R}^{r \times r}$. Here, $\mathbf{X} = \mathbf{E_s} + \mathbf{B}_{:r}$ sums token and positional embeddings $\mathbf{B} \in \mathbb{R}^{n \times d}$. Positional embeddings are independent from the observed context $\mathbf{s}$. After attending to context $\mathbf{s}$, token embeddings are *unembedded* by taking inner products with each token embedding vector $\mathbf{E}_i$. Logits are viewed as a $n \times t$ matrix. Each row contains, on its $i$-th column, the inner product of $\mathbf{E}_i$ with the $d$ dimensional attended context vector. Logits' $r$-th row writes

$$\mathtt{logits}(\mathbf{s})_{r,:} = \{\mathtt{att}(\mathbf{E_s} + \mathbf{B})\}_{r,:}(\mathbf{E_s} + \mathbf{B}_{:r})\mathbf{V}\mathbf{E}^\mathsf{T} .$$

The next token $\hat{\mathbf{s}}_{r+1}$ is drawn from the multinomial distribution parameterized by logits' softmax:

$$\hat{\mathbf{s}}_{r+1} \sim \mathbb{P}[\mathbf{s}_{r+1}|\mathbf{s}_{:r}] = \mathtt{softmax}\{\mathtt{logits}(\mathbf{s})_{r,:}\} . \tag{1}$$

Parameters are optimized via the cross-entropy loss $\ell$ summed over prefixes of sequences in the training set:

$$\mathcal{L} = \sum_{\mathbf{s}} \sum_{r \in [n]} \ell\left(\mathtt{softmax}\{\mathtt{logits}(\mathbf{s})_{r,:}\}, \delta_{\mathbf{s}_{r+1}}\right) , \tag{2}$$

where $\delta_i$ is the one-hot indicator vector for token $i$: $\delta_i = [0, \cdots, 0, \underset{i}{1}, 0, \cdots, 0]^\mathsf{T} \in \mathbb{R}^t$. We are interested in a single generation round: repeatedly applying Eq. (1) to the initial context stacked with the predicted tokens until a stopping condition is met.

## 3.2 ADDED POSITIONAL EMBEDDINGS AUTOREGRESS, HIGH DIMENSIONS MEMORIZE

We analyze sequence generation by a single attention head when Eq. (1) is applied recursively. Our goal in this theoretical study is to demonstrate the existence of a Transformer model capable of depth extrapolation. For this, it is sufficient to prove that a simplified model fulfills the property. Hence, we will restrict our study to a single sequence loss, which will provide an extrapolator as demonstrated in Theorem 4.1, and we make two simplifying assumptions. (1) We use uniform attention $\mathbf{Q} = 0$. For sequences of fixed length with pre-aligned contexts, the use of the attention weights for re-alignment of sequences is not required. For example, we will see in Section 4 that to complete parentheses sequences, we only need to access partial sums of sequence tokens. This is performed by uniform attention. (2) We use a value matrix proportional to the identity $\mathbf{V} = v\mathbf{I}_d$, to isolate the contribution of positional embeddings to autoregression. We prove that a global optimizer of $\mathcal{L}$ with a single term

s belongs to this subset. One such attention head with added positional embeddings can perfectly fit to a single sequence.

Under these conditions, and assuming $\mathbf{E}\mathbf{E}^\top \approx \mathbf{I}_t$, the first-order condition on the loss $\mathcal{L}$ in Eq. (2) simplifies to a system of equations. For each $r \in [n]$:

$$\texttt{softmax}\left(vr^{-1}\mathbf{1}_r^\top\{\mathbf{E}_\mathbf{s} + \mathbf{B}_{:r}\}\mathbf{E}^\top\right) - \delta_{\mathbf{s}_{r+1}}^\top \approx 0 \ . \tag{3}$$

This simplification facilitates derivation of the positional embeddings $\mathbf{B} = \mathbf{B}(\mathbf{s}) \in \mathbb{R}^{n\times d}$ from Eq. (3), using telescoping partial sums, as a recursive relation using a free parameter $\gamma \in \mathbb{R}$ (calculation detailed in Appendix 5.2):

$$\begin{cases} \mathbf{B}_1 = -\mathbf{E}_{\mathbf{s}_1} + \gamma\mathbf{E}_{\mathbf{s}_2} \\ \mathbf{B}_i = -\mathbf{E}_{\mathbf{s}_i} + \gamma(\mathbf{E}_{\mathbf{s}_{i+1}} - \mathbf{E}_{\mathbf{s}_i}) \ \text{for} \ i \geq 2 \ . \end{cases} \tag{4}$$

The following result explains memorization in the simplified self-attention model we considered. In virtue of the Johnson-Lindenstrauss Lemma (Dasgupta and Gupta, 2003), it allow to draw connections between the context length $n$, vocabulary size $t$, and embedding dimension $d$.

**Theorem 3.1.** *Let $\varepsilon \in (0,1)$, and $\mathbf{s} \in [t]^n$ be a tuple. For $d \geq 4\log n/(\varepsilon^2/2 - \varepsilon^3/3)$, there exists a token embedding $\mathbf{E} \in \mathbb{R}^{t\times d}$ such that a self-attention model as in Eq. (1), with $\mathbf{B}$ defined as in Eq. (4), given any prefix of $\mathbf{s}$ as context, exactly completes $\hat{\mathbf{s}} = \mathbf{s}$ with probability at least $1 - nt\exp\{-v\gamma/n + 2\varepsilon\}$. The next-token prediction cross-entropy loss value of the model is therefore lower than $2n^2t\exp\{-v\gamma/n + 2\varepsilon\}$, which decays to zero for large enough $\gamma v$. $\mathbf{E}$ can be computed in polynomial time.*

This result holds useful insights that can guide our thinking beyond the set of assumptions.

**Scaling.** In the probability expression above, we can substitute $\varepsilon = 2\left(\{2/d\}\log n\right)^{1/2}$. This relation shows how the correct completion probability increases with large embedding dimension:

$$\mathbb{P}\left[\hat{\mathbf{s}} = \mathbf{s}\right] \geq 1 - nte^{-v\gamma/n}(1 + 4\sqrt{(2\log n)/d}) \ .$$

**Memorization.** In summary, we stated that when the mixed contribution of the value operator $v$ and the added positional residual term $\gamma$ which appear in the product $v\gamma$ gets large, then the attention head fits the observed sequence and is able to re-generate it by repeated applications of Eq. (1). Notably, $v\gamma \gg 1$ if both of $v$ and $\gamma$ have the same sign, for bounded $|\gamma|$ and $|v| \gg 1$.

**Stability.** If $\gamma = -1/2$ (and $-v \gg 1$), the model is attractive: it completes other contexts with a sequence of tokens that minimize distance to $\mathbf{E}_\mathbf{s}$. In contrast, $\gamma = 1/2$ produces a repulsive model which pushes completions of different prefixes away from $\mathbf{E}_\mathbf{s}$. To build an intuition on stability, consider a sequence $\mathbf{z} \in [t]^n$ generated by the same model, and let $c_r = \mathbf{E}_{\mathbf{s}_r} - \mathbf{E}_{\mathbf{z}_r}$. Assuming for all $r = 1 \cdots n$,

$$\frac{v}{r}\mathbf{1}_r\left(\mathbf{E}_{\mathbf{s}_{:r}} + \mathbf{B}_{:r}\right) = v\gamma\mathbf{E}_{\mathbf{s}_{r+1}} \ \text{and} \ \frac{v}{r}\mathbf{1}_r\left(\mathbf{E}_{\mathbf{z}_{:r}} + \mathbf{B}_{:r}\right) = v\gamma\mathbf{E}_{\mathbf{z}_{r+1}} \ ,$$

we get $c_{r+1} = \mu_r c_r$ with $\mu_r = (1 + [r+1]\gamma)/(r\gamma)$. If $\gamma = -1/2$, $\mu_r = 1 - 1/r < 1$ resulting in a contractive sequence of differences $c_r$ and if $\gamma = 1/2$, $\mu_r = 1 + 3/r > 1$ and the sequence $c_r$ diverges.

**Multiple sequences.** To expand the stated result to more than one sequence in all generality, it is necessary to formulate the problem as an autoregression in $\mathbf{V}$. $\mathbf{V}$ is not a multiple of identity in this situation. Matrices $\texttt{att}$ and $\mathbf{B}$ bound the condition number of the regression. A sufficient condition for perfectly fitting (interpolating) $m$ sequences $\mathbf{s}^{(1)} \cdots \mathbf{s}^{(m)}$ is the existence of $\mathbf{B}$ such that $\mathbf{E}_{\mathbf{s}_i^{(j)}} + \mathbf{B}_i$'s form an orthogonal set. This suggests $nm$ upper bounds $d$. The decomposition

$$\texttt{att}_\mathbf{z}(\mathbf{E}_\mathbf{z} + \mathbf{B})\mathbf{V} = \gamma v\mathbf{E}_{\mathbf{s}_{r+1}^{(j)}} + \{\texttt{att}_\mathbf{z} - \texttt{att}_{\mathbf{s}^{(j)}}\}(\mathbf{E}_\mathbf{z} + \mathbf{B})\mathbf{V} + \texttt{att}_{\mathbf{s}^{(j)}}\{\mathbf{E}_\mathbf{z} - \mathbf{E}_{\mathbf{s}^{(j)}}\}\mathbf{V}$$

where $\texttt{att}_\mathbf{s} = \texttt{att}(\mathbf{E}_\mathbf{s} + \mathbf{B})$, suggests that $\mathbf{z}$'s completion follows the closest $\mathbf{s}^{(j)}$. Discussing relative positions of sequences $\mathbf{s}^{(j)}$ and assumptions on $\mathbf{V}, \mathbf{B}, \texttt{att}$ hold keys to characterizing the multi-sequence problem which we defer to future work.

## 4 GENERATION OF A BOUNDED DYCK LANGUAGE WITH A TRANSFORMER

In this section we apply our theoretical findings and confront them with numerical experiments on the simplest example of nested structure. Bounded Dyck words, denoted as $\mathcal{D}_{2N}$, comprise all

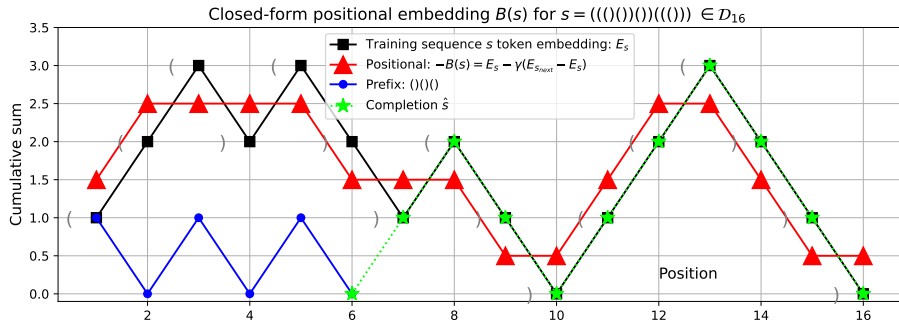

Figure 2: Single layer attention $\gamma = -1/2, -v \gg 1$: closed-form positional embeddings (red) and completion (lime) of a prefix (blue) for the training sequence **s** (black) are represented as paths in the space of cumulative token embeddings.

strings of length $2N$ of balanced parentheses. Dyck words are studied in computational linguistics for their minimalistic representation of recursion and hierarchy in a formal language (Chomsky and Schützenberger, 1959). Prior work studied full Transformers that learn to recognize $\mathcal{D}_{2N}$ sequences Ebrahimi et al. (2020); Yao et al. (2021); Delétang et al. (2023); Murty et al. (2023). Our focus in the theoretical section is on a smaller set of models which we demonstrate are sufficient to solve the problem. In theory and experiments we are interested in sequence generation in a single pass. We are particularly interested in studying depth extrapolation. The depth of a Dyck word is the maximum parenthesis stack size or largest cumulative sum: highest point of the lattice as in Figure 2.

The Catalan number enumerates the set of balanced parentheses of length $2N$: $C_N = |\mathcal{D}_{2N}| = \binom{2N}{N}/(N+1) \simeq 4^N \pi^{-1/2} N^{-3/2}$, see Roman (2015) for a comprehensive reference. Given a short prefix, the number of valid correct completions is exponentially large. Therefore, the autoregressive loss of Eq. (2) must be fed an exponentially large number of data-points in order to learn to complete any given prefix to any of the language's full sequences. In that scenario, ambiguity on the correct completion leads to difficulty in evaluation for this ill-posed problem. However, if our goal is not to complete a prefix to *the correct sequence*, but rather to complete the prefix into *a valid sequence*, then the problem is well defined. Sticking to this convention, we will show that minimizing the autoregressive loss in Eq. (2) on a single sample (any of the $\mathcal{D}_{2N}$ sequences) and in dimension one, yields a global optimum which solves the problem with desirably high probability. We establish the closed-form expression of this solution. However, our experiments show that the same loss function, minimized with gradient-descent over a large training set, does not lead to learning the single point abstraction that solves the bigger problem. The large scale of data demands large embedding dimension (reminiscent of Theorem 3.1). But since the number of sequences in the training set is close to $C_N$ and grows exponentially, this approach cannot lead to a viable solution for large $N$. Our numerical experiments also show that the models trained with gradient descent on small training data with only a few sequences fail at identifying the well-suited closed-form model of Theorem 4.1.

## 4.1 Closed-form self-attention models that complete parentheses prefixes

We construct self-attention models with positional embeddings as in Eq. (4), using a deterministic token embedding in a single dimension $d = 1$. We follow the construction used in Theorem 3.1 but with deterministic embeddings. Our embedding constructing is based on the observation (see Roman (2015)) that Dyck words can be represented as paths above the $y = 0$ line in a plane with integer coordinates where the $x$ axis represents positions and the $y$ axis at position $x$ denotes the size of the open parenthesis stack. This simple geometric interpretation guides our choice of embedding: '(' $\rightarrow$ 1 and ')' $\rightarrow$ -1. Their partial cumulative sums provided by the uniform attention trace the path across position indices. We construct positional embeddings following Eq. (4) on $\mathcal{D}_{2N}$, with $\gamma = -1/2$ and $\mathbf{E} = [1, \quad -1]^\top$. We visualize this construction in Figure 2 a $\mathcal{D}_{16}$ sequence (black squares), the

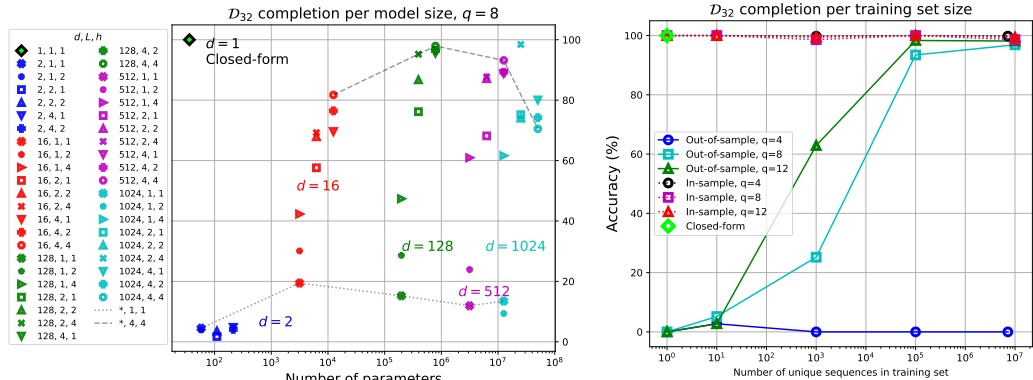

Figure 3: Left: empirical completion rates of learned and closed form expression $\mathcal{D}_{32}$ generators, as a function of number of parameters. Note the black diamond on the upper left corner, which represents the high performing closed-form single layer model with $d = 1$. Right: performance of Transformers in out-of-sample (full lines) and in-sample (dashed lines) data, trained on subsets of $\mathcal{D}_{32}$ with varying number of unique sequences and depth $q$. Again, the upper left corner lime diamond shows the superior performance of a closed-form single head model which gradient descent fails at identifying.

associated positional embedding (red triangles) and an example of context or prefix (blue dots) and completion (dashed lime stars).

**Theorem 4.1** (Closed-form expression attention head for $\mathcal{D}_{2N}$). *Take an integer $N > 0$ and consider any $\mathbf{s} \in \mathcal{D}_{2N}$ as the single training sequence. The one-dimensional token embedding $\mathbf{E} = \begin{bmatrix} 1, & -1 \end{bmatrix}^{\mathsf{T}}$ and the positional embedding $\mathbf{B} = \mathbf{B}(\mathbf{s}) \in \mathbb{R}^{2N \times 1}$ defined as in Eq. 4, form a self-attention model with exactly $2N + 3$ weights achieving a cross-entropy loss lower than $8N^2 \exp(-N)$.*

- ***Memorization**. If $v\gamma > N^2$, a single generation pass over Eq. (1) completes any prefix $\mathbf{s}_{:r}$ of $\mathbf{s}$ into $\mathbf{s}$ with probability at least $1 - 2N \exp(-N)$.*

- ***Generalization**. With $\gamma = -1/2$ and $-v > 2N^2$, the model completes prefixes $\mathbf{z}_{:r}$ of any sequence $\mathbf{z} \in \mathcal{D}_{2N}$ with arbitrary depth, into a complete $\mathcal{D}_{2N}$ sequence (balanced parentheses) with probability at least $1 - 2N \exp(-N)$. Conversely, if $\gamma > 0$, then if the prefix $\mathbf{z}_{:r}$ has a different stack size than $\mathbf{s}$'s prefix of the same size $\mathbf{1}_r^{\mathsf{T}} \mathbf{E}_{\mathbf{z}_{:r}} \neq \mathbf{1}_r^{\mathsf{T}} \mathbf{E}_{\mathbf{s}_{:r}}$, then even with the condition $v\gamma > N^2$ required for memorization, the model will fail at generating valid Dyck sequences with high probability.*

Our results provide a more compact and precise solution than the two-layer Transformers by Yao et al. (2021), due to learned positional embeddings instead of hard-coded linear embeddings. The result shows that completion of $\mathbf{z}_{:r}$ follows the shortest path in the space of stack sizes paths representing the parentheses stacks, until it overlaps with the cumulative sum of the training sequence $\mathbf{s}$. In contrast, the model with $v > 0$ will amplify a difference in stack sizes of the prefixes. We call out that the $N \exp(-N)$ terms (smaller than $10^{-3}$ for $N \geq 10$) in the probability lower bounds of this section can be adjusted with a free parameter in the exponential for further reduction if needed. However, to be concise we have opted for this notation.

### 4.2 NUMERICALLY LEARNING A BOUNDED DYCK LANGUAGE

Our numerical experiments offer a deeper understanding of the capabilities and limitations of Transformers at depth generalization, on Dyck language. The role of embedding dimension, number of layer and heads are first examined. We then study sample complexity. We numerically compare the performance of the closed-form solution with the trained models on deeper out-of-sample test sets.
**Data**. Our training and test data are sampled from $\mathcal{D}_{32}$: balanced parentheses words of length 32. Using the Catalan numbers (Roman, 2015) $C_N$, we enumerate $C_{16} = \binom{32}{16}/17 \approx 3.5 \cdot 10^7$ such

words. A set of $2^{25} \approx 3.4 \cdot 10^7$ randomly generated words is split into training and test sets, using depth of the parentheses expression as the split criterion. The training data has $\approx 2.4 \cdot 10^7$ samples of depth $q \leq 8$, and we report accuracy (the number of completion of prefixes into balances parentheses sequences) on a random subset of 1024 distinct sequences from the test data where prefixes have depth $q \geq 9$. Our completion criterion consists of running the `generate` method to get a string of $2N = 32$ characters. We then examine whether the parentheses sequence is balanced. We report accuracy numbers evaluated on a sample of 1024 test data. Our depth-based train-test split allows to measure depth extrapolation. It distinguishes our work from Murty et al. (2023)'s work where uniform split of data was employed to measure generalization within the same depth.

**Models**. We train decoder Transformer models (similar in architecture to GPT-2, adapted from `nanoGPT` (Karpathy, 2022)) with varying $L = 1, 2, 4$ layers, number of heads $h = 1, 2, 4$ and embedding dimensions of $d = 2, 16, 128, 512, 1024$: all combinations where $h$ divides $d$ are tested. In the Transformer model, each attention block contains a directed self-attention, which is followed by a multi-layer perceptron (MLP) with 2-layers and $8d$ neurons in the hidden layer. For training the models, we use minibatches of size 8, AdamW with learning rates between $6 \cdot 10^{-6}$ to $6 \cdot 10^{-5}$ runs for a maximum of 10,000 iterations and a dropout of 0.1.

**Observations.** Figure 3 (left) shows that models' best performance grows with embedding dimension, indicating its first order influence on accuracy. Large embedding models necessitate multiple layers and attention heads to perform well. From the performance decay of the $h = L = 1$ models (connected with a dotted line), we observe that the importance of $L$ and $h$ increases with larger $d$. Models with $d \geq 128$ and $h \geq 2, L = 4$ hit $95 + \%$ accuracy. Conversely, gradient trained $d = 2$ models all perform poorly. Notably, the only model which reaches $100\%$ accuracy is the closed-form model with $d = h = L = 1$, represented in our plot as a lime-color diamond with a black contour on the upper left corner.

**Sample size experiment.** In a subsequent experiment, we examine how sample complexity is impacted by depth. For this, we train the largest model with $d = 1024, L = h = 4$ on training sets containing different numbers of distinct sequences. We select $10^k$ random sequences for $k = 0, 1, 3, 5, 7$ from a training set with expressions of depth $q \leq 4, 8, 12$. Our test set consists of 1024 prefixes of depth at least $q \geq 13$. We evaluate each of the obtained models at prefix completion on the training set itself, to evaluate in-sample performance. These are represented as dotted lines in Figure 3 right. They all appear on top of the plot as all models fit well to data and perform well at in-sample completion of prefixes. We also evaluate completion accuracy on a subset of 1024 distinct prefixes of depth at least $q \geq 13$ from the test set, to measure out-of-sample performance. These results are shown in the right plot in Figure 3 as full lines.

We observe that the gap between the dashed (in-sample accuracy) and the full lines (out-of-sample accuracy) for small values of the training set size are large. This suggests that the most successful large and greedy model perform well when they fit to a large training set. They do not learn functions similar to our closed-form model presented in Section 4.1 on small samples. We empirically evaluate the closed-form model and record a $100\%$ accuracy on this benchmark. We represent this in Figure 3 right as a lime colored diamond on the upper left corner (single sequence for training). It is noteworthy that the model trained on shallow sequences of depth $q = 4$ has very low out-of-sample accuracy on the deep test data, even when the number of training samples is large. This suggests that generalization to deeper expressions is harder as the gap between training data depth and test data depth widens. For all trained models with large enough embedding dimension, the training loss value of a Transformer model trained on a single sequence reaches machine precision's zero. However the models seem to identify a global minimum which does not have the desired property of the closed-form model. This observation suggests that even with the right implicit bias in model architecture and training data selection, gradient descent fails to find the desired optimum of the objective function.

**Completion distributions for two models.** To deepen our intuition on sequence generation behaviors of the trained models, we study generated sequences by 2 models: (1) the model trained with a single $q = 4$ sample '(()((()))((()()()))()(()))(()())' and (2) the model trained on $10^7$ distinct sequences $q \leq 8$ which appears to generalize better. We examine the distribution of generated sequences when we pass in the same context to each model in 100 independent generation trials. We observe, see Figure 4, that (top left plot) when a prefix of the training sample is passed as input to the single sequence model, it completes it 100 times out of 100 into the training sample and hence achieves an accuracy of 100 / 100. However, the same model fails at completing deeper (top middle)

or shallower (top right) contexts. Interestingly, in both scenarios the model repeats a pattern similar to the training sample which it has seemingly memorized. In the case of the deep context, this results in an open stack completion. In the case of the shallow context, the generated sequence is invalid as it contains more closed than open parentheses. The model trained on $10^7$ samples generates a more diverse set of sequences, yet the generated sequences for a $q = 9$ deep sequence (lower middle plot) seem to show a lower accuracy. Finally, we evaluate the two models 100 times on the deepest context of $\mathcal{D}_{32}$ with 16 open parentheses. We observe that the large sample model correctly completes the context 52 out of 100 times, while the single sequence model fails at generating the right completion (see Table 1 in Appendix 5.3.2).

**Spectral analysis.** We examine the spectrum of the value matrix (times projection transpose), numerically obtained after training Transformers with 1 and $10^7$ samples. We observe a low-rank structure for the best model (see Figure 5 in Appendix 5.3.3). Large spectral elements pop out when the model is trained on a large training set and is able to generalize, with a dominant negative eigenvalue. This phenomenon was also reported in numerical studies by Trockman and Kolter (2023).

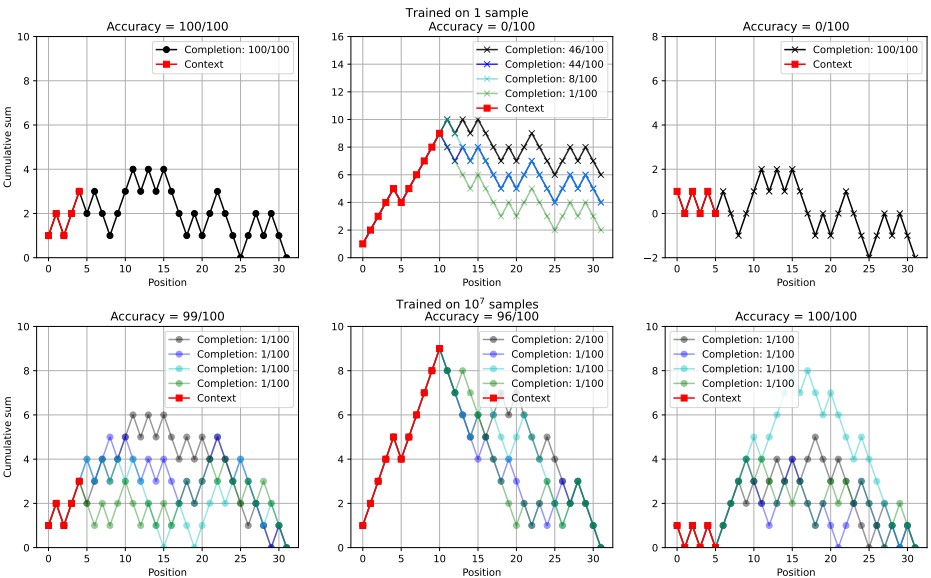

Figure 4: Completions of a few contexts by the model trained on 1 and $10^7$ sequence training sets. Context are chosen as a prefix of the single sequence training data (left), a deeper than training data: depth $q = 9$ (middle) and a shallow but not a prefix of the single sequence training (right). The top row represent completions by the model trained a single $q = 4$ sequence. The bottom row are completions by models trained on $10^7$ samples of depth $q \leq 8$.

## 5 DISCUSSION

To study reasoning on harder than training set tasks we conducted a controlled experiment on simplifying Boolean expressions. We stratified the data by depth of expressions. Evaluations on deep expressions show that a specialized model can overfit the data but struggles at extrapolating to deeper nested expressions. We showed on a simpler problem (Dyck completion) that there exists Transformers capable of depth generalization. Yet, gradient-based pre-training seems to fail at identifying them. Our findings raise further research questions. First, can the closed-form solution be expanded and its generalization power discussed when it interpolates multiple sequences of bounded lengths? Can it cover other bracket types, Boolean logic, propositional, fuzzy and temporal logic? Second, the characterization of the more stable solutions which generalize in addition to interpolating can hold keys to biasing gradient trained models to reach such solutions. Finally, if extrapolation in a single pass appears to be out-of-reach, how do we optimize cost of training models on larger and deeper synthetic training sets, versus calling inference routines multiple times?

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

APPENDIX

## 5.1 BOOLEAN EXPRESSION SIMPLIFICATION

### 5.1.1 LLM EXAMPLE

GPT-4 returns correct simplifications for 9 out of 10 samples for simplifying logical expressions with 2 clauses of depth 1. However, with 5 clauses of depths $q = 3$, the model correctly accomplishes 0 out of 10 tasks. Example: Can you simplify the following Boolean expression?
$(((\neg p2|\neg p2)|(false|(false\&false))))\&(((false\&true)|((p3|p3)\&(\neg p3|\neg p2))))$
$\&(((p2|\neg p2)\&((\neg p0\&\neg p1)|(p3|\neg p2))))|(((\neg p0|(p1\&\neg p3))|(p2\&(p2|p3))))$
$\&(((\neg p1|p1)\&(p1|false)))$

---

GPT-4: To simplify the provided boolean expression, we'll analyze and simplify each subexpression and then combine them into a final simplified form:
*Original Expression:*

$(((\neg p2|\neg p2)|(false|(false\&false))))\&(((false\&true)|((p3|p3)\&(\neg p3|\neg p2))))$

$\&(((p2|\neg p2)\&((\neg p0\&\neg p1)|(p3|\neg p2))))|(((\neg p0|(p1\&\neg p3))|(p2\&(p2|p3))))$

$\&(((\neg p1|p1)\&(p1|false)))$

Simplification Steps:

1. **Simplify subexpressions:**

   - $\neg p2|\neg p2$ simplifies to $\neg p2$.
   - `false`$|$(`false`&`false`) simplifies to `false`.
   - `false`&`true` simplifies to `false`.
   - p3$|$p3 simplifies to p3.
   - p2$|\neg$p2 simplifies to `true` (this is a tautology).
   - $\neg$p1$|$p1 simplifies to `true` (also a tautology).
   - p1$|$`false` simplifies to p1.

2. **Reconstruct expression using simplified components:**

   - The first section simplifies to $\neg$p2&((`false`$|$(p3&($\neg$p3$|\neg$p2)))&`true`)
   - Using the simplifications, p3&($\neg$p3$|\neg$p2) reduces to p3&$\neg$p2.
   - The overall expression becomes:

   $$(\neg p2\&(p3\&\neg p2))|((\neg p0|(p1\&\neg p3))|(p2\&(p2|p3)))\&true\&p1$$

   - p2&(p2$|$p3) simplifies to p2.

3. **Further combine and simplify:**

   - $\neg$p2&(p3&$\neg$p2) simplifies to p3&$\neg$p2 (since $\neg$p2&$\neg$p2 is just $\neg$p2).
   - Combine the remaining parts:

   $$(p3\&\neg p2)|((\neg p0|(p1\&\neg p3))|p2)\&true\&p1$$

   - Since `true`&p1 is just p1, simplify to:

   $$(p3\&\neg p2)|((\neg p0|p1)|p2)\&p1$$

   - Further simplification yields:

   $$((p3\&\neg p2)|\neg p0|p2)\&p1$$

*Final Simplified Expression:*$((p3\&\neg p2)|\neg p0|p2)\&p1$. This is the simplified version of the complex boolean expression you provided. Let me know if you need any further clarification on the steps!

**Note:** The correct solution is p1|(p3&¬p2)

### 5.1.2 RANDOM BOOLEAN EXPRESSION GENERATOR

We stack a list of clauses with random joins defined as

$$\texttt{random\_join}\,(\texttt{a}, \texttt{b}) = \begin{cases} \texttt{a \& b} & \text{with probability } 1/2 \\ \texttt{a | b} & \text{with probability } 1/2. \end{cases} \tag{5}$$

We want to generate a random logical clause with $\texttt{true}, \texttt{false}, \texttt{p}_1, \cdots \texttt{p}_3$ where $\texttt{p}_1, \cdots \texttt{p}_3$ are Boolean variables. We generate random logical clauses of a given maximum depth $q$ with recursive calls to a function $\texttt{random\_clause}$ defined on $\texttt{variables} = \{\texttt{true}, \texttt{false}, \texttt{p}_1, \neg\texttt{p}_1 \cdots \texttt{p}_3, \neg\texttt{p}_3\}$ recursively as in Algorithm 5.1.2. Given a desired number of clauses $k$ in an expression, we call $\texttt{random\_clause}$ $k$ times and random join the $k$ expressions, each of which has maximum depth $q$.

---

**Algorithm 1** Random Boolean clause generator

---

1: **function** RANDOM_CLAUSE($q$)
2:     **if** $q \leq 0$ **then**
3:         **return** choice(variables)
4:     **else**
5:         left ← choice([random_clause(q − 1), random_clause(q − 2)])
6:         right ← choice([random_clause(q − 1), random_clause(q − 2)])
7:         **return** random_join[left, right]
8:     **end if**
9: **end function**

---

### 5.2 PROOF OF THEORETICAL RESULTS

We begin with stating a preliminary result where tokens are one-hot encoded.

**Lemma 5.1.** *Let $\mathbf{s} \in [t]^n$ be a sequence of tokens. Under uniform attention $\mathbf{Q} = 0$ and one-hot token embeddings where token $i$ is embedded as $\delta_i$ we use $\mathbf{I}_t = [\delta_1^{\mathsf{T}}, \cdots, \delta_t^{\mathsf{T}}]^{\mathsf{T}}$ as the token embedding matrix. Consider the positional embedding $\mathbf{R} = [\mathbf{R}_1, \quad \cdots \quad , \mathbf{R}_n]^{\mathsf{T}}$, where*

$$\begin{cases} \mathbf{R}_1 = -\delta_{\mathbf{s}_1} + \gamma\delta_{\mathbf{s}_2} \\ \mathbf{R}_i = -(1 + \gamma)\delta_{\mathbf{s}_i} + \gamma\delta_{\mathbf{s}_{i+1}} \text{ for } i \geq 2 . \end{cases} \tag{6}$$

*and $\mathbf{V} = v\mathbf{I}_t$ with $v\gamma > n\log(nt)$. Then for $r < n$ and the 1 model with these parameters, any prefix $\mathbf{s}_{:r}$ of $\mathbf{s}$ is completed into $\mathbf{s}$, i.e. we get $\hat{\mathbf{s}} = \mathbf{s}$, with probability at least $1 - nt\exp(-v\gamma/n)$.*

*Proof of Lemma 5.1.* With uniform attention, the $r$-th row of $\texttt{att}$ is $r^{-1}\mathbf{1}_r^{\mathsf{T}}$. We know that, by construction,

$$\left(\mathbf{1}_r^{\mathsf{T}}\mathbf{R}_{:r,:}\right)^{\mathsf{T}} = \sum_{i=1}^{r}\mathbf{R}_i = -\delta_{s_1} + \gamma\delta_{\mathbf{s}_2}$$

$$-\delta_{s_2} - \gamma\delta_{\mathbf{s}_2} + \gamma\delta_{\mathbf{s}_3}$$

$$-\delta_{s_3} - \gamma\delta_{\mathbf{s}_3} + \gamma\delta_{\mathbf{s}_3}$$

$$\vdots$$

$$= -\left(\mathbf{1}_r^{\mathsf{T}}\delta_{\mathbf{s}_{:r,:}}\right)^{\mathsf{T}} + \gamma\delta_{\mathbf{s}_{r+1}} ,$$

so $\mathbf{1}_r^{\mathsf{T}}\left(\delta_{\mathbf{s}} + \mathbf{R}\right) = \gamma\delta_{\mathbf{s}_{r+1}}^{\mathsf{T}}$ and $\texttt{logits}$ is proportional to the $(r+1)$-th token's one-hot:

$$\texttt{logits}_{r,:} = \frac{v}{r}\mathbf{1}_r^{\mathsf{T}}\left(\delta_{\mathbf{s}} + \mathbf{R}_{:r}\right)\mathbf{I}_t^{\mathsf{T}} = \frac{v\gamma}{r}\delta_{\mathbf{s}_{r+1}}^{\mathsf{T}} .$$

The next token is generated as in 1, from a multinomial drawn from `softmax(logits)`. It follows that next token is $\mathbf{s}_{r+1}$ with probability $[1 + (t-1)\exp(-v\gamma/r)]^{-1} \geq 1 - (t-1)\exp[-v\gamma/n]$ . We assumed that $v\gamma > n\log(nt)$. This means $(n-1)(t-1)\exp[-v\gamma/n] < nt\exp[-v\gamma/n] < 1$, and therefore union bound allows to conclude. $\square$

With this result in hand, we can turn on to our main statement.

*Proof of Theorem 3.1.* We invoke the Johnson-Lindenstrauss Dasgupta and Gupta (2003) Lemma on one-hot embeddings $\{\delta_{\mathbf{s}_i}\}_{i=1}^n$ of Lemma 5.1 to construct $\mathbf{E} \in \mathbb{R}^{t \times d}$ in polynomial time. We use $\delta_{\mathbf{s}_i}^\mathsf{T}\mathbf{E} = \mathbf{E}_{\mathbf{s}_i}$ and write for $i \neq j \in [n]$,

$$(1-\varepsilon)\|\delta_{\mathbf{s}_i} - \delta_{\mathbf{s}_j}\|_2^2 \leq \|\mathbf{E}_{\mathbf{s}_i} - \mathbf{E}_{\mathbf{s}_j}\|_2^2 \leq (1+\varepsilon)\|\delta_{\mathbf{s}_i} - \delta_{\mathbf{s}_j}\|_2^2 \ . \tag{7}$$

We use $\mathbf{V} = v\mathbf{I}_d$ and superscript $\mathbf{E}$ to denote $\texttt{logits}^\mathbf{E}$ calculated with $\mathbf{E}$. We build positional embeddings using a multiplication of $\mathbf{R}$ by the same matrix $\mathbf{E}$ as $\mathbf{B} = \mathbf{R}\mathbf{E} \in \mathbb{R}^{n \times d}$. We have

$$\texttt{logits}^\mathbf{E}_{r,:} = \frac{v}{r}\mathbf{1}_r^\mathsf{T}(\mathbf{E}_\mathbf{s} + \mathbf{B})\mathbf{E}^\mathsf{T} = \frac{v}{r}\mathbf{1}_r^\mathsf{T}(\delta_\mathbf{s} + \mathbf{R})\mathbf{E}\mathbf{E}^\mathsf{T}$$
$$= \frac{v\gamma}{r}\delta_{\mathbf{s}_{r+1}}^\mathsf{T}\mathbf{E}\mathbf{E}^\mathsf{T} \tag{8}$$

We use the Lipschitz property of the linear map $\mathbf{E}$ within the span of $\{\delta_{\mathbf{s}_i}\}_{i=1}^n$ and use its singular value decomposition to establish that $\Xi = \mathbf{E}\mathbf{E}^\mathsf{T}$ is a positive semi-definite matrix with restricted eigenvalues in the span of $\{\delta_{\mathbf{s}_i}\}_{i=1}^n$ lay within $1-\varepsilon$ to $1+\varepsilon$. It therefore satisfies, within the span of $\{\delta_{\mathbf{s}_i}\}$, $\|\mathbf{x}\,\Xi - \mathbf{x}\|_2 \leq \varepsilon$ . As a consequence, within the span of $\{\delta_{\mathbf{s}_i}\}$, we also get, for a row vector $\mathbf{x} \in \mathbb{R}^{1 \times t}$ $\|\mathbf{x}\,\Xi - \mathbf{x}\|_\infty \leq \varepsilon$. This implies that

$$\texttt{softmax}(\texttt{logits}^\mathbf{E}_{r,:})_{\mathbf{s}_{r+1}} \geq \frac{\exp(\frac{v\gamma}{r} - \varepsilon)}{\exp(\frac{v\gamma}{r} - \varepsilon) + (t-1)\exp(\varepsilon)}$$
$$\geq 1 - t\exp\{-v\gamma/n + 2\varepsilon\} \ .$$

We use the union bound to establish that the $d$-dimensional self-attention model generates $\hat{\mathbf{s}} = \mathbf{s}$ with probability at least $1 - nt\exp\{-v\gamma/n + 2\varepsilon\}$. In order to upper bound the loss function, note that each of the $n$ terms in the loss function is independently derived from the probability upper bound we just established:

$$\mathcal{L} = \sum_{r \in [n]} \ell(\texttt{softmax}\{\texttt{logits}(\mathbf{s})_{r,:}\}, \delta_{\mathbf{s}_{r+1}})$$

$$= \sum_{r=0}^{n-1} \ell(\mathbb{P}[\mathbf{s}_{r+1}|\mathbf{s}_{:r}], \delta_{\mathbf{s}_{r+1}})$$

$$= -\sum_{r=0}^{n-1} \log \mathbb{P}[\mathbf{s}_{r+1}|\mathbf{s}_{:r}]$$

$$\leq -\sum_{r=0}^{n-1} \log\left(1 - nt\exp\{-v\gamma/n + 2\varepsilon\}\right) \quad \text{(probability upper bound)}$$

$$\leq 2n^2 t\exp\{-v\gamma/n + 2\varepsilon\} \quad\quad (\text{using } -\log(1-x) \leq 2x \text{ for large enough } \gamma) \ .$$

$\square$

## 5.3 Generation of a bounded Dyck language

### 5.3.1 Proof of theoretical results

Let us start with a useful observation:

**Lemma 5.2.** *For a two-word token embedding* $\mathbf{E} = [1, \quad -1]^\mathsf{T}$, *the logits simplify to* $\mathtt{logits} = [X, -X]$. *Consequently, if* $X > 0$, *the next character generated is an open parenthesis '(' with probability at least* $1 - \exp(-2X)$, *if* $X < 0$, *a closed parenthesis ')' is generated with probability at least* $1 - \exp(2X)$.

*Proof of Lemma 5.2.* We know that $\mathtt{logits} = X\mathbf{E}^\mathsf{T} = [X, -X]$, which means that probabilities are given by

$$[\mathbb{P} \text{ next character is '(' }, \mathbb{P} \text{ next character is ')'}] = \mathtt{softmax}([X, -X])$$

$$= \left[ \frac{e^X}{e^X + e^{-X}}, \frac{e^{-X}}{e^{-X} + e^X} \right] \quad,$$

and if $X > 0$, the inequality $1/(1 + e^{-2X}) \geq 1 - e^{-2X}$ proves the result. $\qquad\square$

Using this, we can turn on to the proof of the main result.

*Proof of Theorem 4.1.* Consider the vocabulary '(',')' tokenized as '(' $\to 0$, ')' $\to 1$, i.e. $\mathbf{E} = \begin{bmatrix} 1 \\ -1 \end{bmatrix}$ and uniform attention, i.e. $\mathbf{Q} = 0$. With this choice the $r$-th row of $\mathtt{att}\ \mathbf{E_s}$ is equal to the size of the stack at entry $r$, $\Delta_r$ defined as

$$\Delta_r(\mathbf{s}) = |\ i \leq r\ ,\ \mathbf{s}_i = \text{'('}\ | - |\ i \leq r\ ,\ \mathbf{s}_i = \text{')'}\ | = \mathbf{1}_r^\mathsf{T}\mathbf{E}_{\mathbf{s}_{:r}}\quad. \tag{9}$$

It is the size of the *stack* among the first $r \geq 1$ characters, that is the number of open parenthesis '(' minus number of closed parentheses ')' among the first $r$ characters. We will use $\mathbf{B}$ defined as Eq. (4) with $\gamma = -1/2$ and $v < -2N^2$. We want to prove two results. **Memorization**: a prefix of $\mathbf{s}$ is completed into $\mathbf{s}$ and **generalization**: any other prefix is completed with valid $\mathcal{D}_{2N}$ tokens.

**Memorization**. We use Eq. 8 to write $\mathtt{logits}_{r:} = (v\gamma/r)\delta_{\mathbf{s}_{r+1}}^\mathsf{T}\mathbf{E}\mathbf{E}^\mathsf{T}$ and use $\mathbf{E}$'s expression to write:

$$\mathtt{logits}_{r,:} = \frac{v\gamma}{r}\delta_{\mathbf{s}_{r+1}}^\mathsf{T}\mathbf{E}\mathbf{E}^\mathsf{T} = \begin{cases} \frac{v\gamma}{r}\mathbf{E}^\mathsf{T} & \text{if } \mathbf{s}_{r+1} = \text{'('} \\ -\frac{v\gamma}{r}\mathbf{E}^\mathsf{T} & \text{if } \mathbf{s}_{r+1} = \text{')'} \end{cases}\quad.$$

This is because we get $\delta_{\mathbf{s}_{r+1}}^\mathsf{T}\mathbf{E} = 1$ if $\mathbf{s}_{r+1} = \text{'('}$ and $-1$ otherwise. Using the result of Lemma 5.2 we have proven that with our choice of $\gamma v > N^2$, with probability at least $1 - \exp(-\gamma v/r) \geq 1 - \exp(-N)$, the next character of the generated sequence is $\mathbf{s}_{r+1}$, and using union bound, we conclude that with probability at least $1 - 2N\exp(-N)$, each prefix of $\mathbf{s}$ is completed into $\mathbf{s}$. This completes our proof of the memorization result.

**Generalization**. We need to prove that any other sequence $\mathbf{z}$ is completed with a valid parenthesis sequence. Recall that $\mathtt{logits}_r = (v/r)\mathbf{1}_r^\mathsf{T}(\mathbf{E_z} + \mathbf{B}(\mathbf{s})) = [X, -X]$ and

$$\mathbf{1}_r^\mathsf{T}(\mathbf{E_z} + \mathbf{B}(\mathbf{s})) = \sum_{i=1}^r \mathbf{E}_{\mathbf{z}_i} + \mathbf{B}_i(\mathbf{s})$$

$$= \Delta_r(\mathbf{z}) - \Delta_r(\mathbf{s}) + \gamma\mathbf{E}_{\mathbf{s}_{r+1}}$$

Remark that if $\gamma = -1/2$ and $-v > 2N^2$, then the model is elastic: generated next tokens of $\mathbf{z}$ attract the stack size or cumulative sum of $\mathbf{z}$ to that of the training data point $\mathbf{s}$. To prove this, note that if at position $r$, the stack in $\mathbf{z}$ is larger than the stack in $\mathbf{s}$, then because $v < 0$, then $X = v\{\Delta_r(\mathbf{z}) - \Delta_r(\mathbf{s}) + \gamma\mathbf{E}_{\mathbf{s}_{r+1}}\} \leq v\{1 - (1/2)\mathbf{E}_{\mathbf{s}_{r+1}}\}$ which implies that $X < 0$ consequently the next character generated for $\mathbf{z}$ is ')' which reduces the stack of $\mathbf{z}$ one step closer to $\mathbf{s}$. Conversely, if the stack of $\mathbf{z}$ is below that of $\mathbf{s}$, then the sign of $X$ flips and we get an open parenthesis as the next character.

In order to ensure that such a sequence is a valid parenthesis sequence, first look at $r$ such that $\Delta_r(\mathbf{z}) = 0$, i.e. a well-balanced prefix $\mathbf{z}_{:r}$. The generator would violate the parentheses balances if

the next character were to be a closed parenthesis ')'. We are interested in the value of the `logits` at row $r$. With our choice of $\gamma = -1/2$,

$$
\begin{aligned}
\mathbf{1}_r^\top \left( \mathbf{E_z} + \mathbf{B(s)} \right) &= \sum_{i=1}^{r} \mathbf{E_{z}}_i + \mathbf{B}_i(\mathbf{s}) \\
&= \Delta_r(\mathbf{z}) - \Delta_r(\mathbf{s}) + \gamma \mathbf{E_{s}}_{r+1} = -\Delta_r(\mathbf{s}) + \gamma \mathbf{E_{s}}_{r+1} \\
&\leq -\frac{1}{2} < 0 \ .
\end{aligned}
$$

Using Lemma 5.2, with probability at least $1 - \exp(v/r) \geq 1 - \exp(-N)$ the next character generated is an open parenthesis: this ensures that the parentheses stack size is positive and the generator is not violating the constraint. The other violation case is where $\mathbf{z}$ has reached the maximum value of $\Delta_r(\mathbf{z})$: $r > N$ and $\Delta_r(\mathbf{z}) = 2N - r$. In this case, for any sequence $\mathbf{s}$, $\Delta_r(\mathbf{z}) - \Delta_r(\mathbf{s}) \geq 0$. If $\Delta_r(\mathbf{z}) = \Delta_r(\mathbf{s})$, then $\mathbf{s}_{r+1} = $ ')' and this proves that $\mathbf{1}_r^\top \left( \mathbf{E_z} + \mathbf{B(s)} \right) = 1/2$. If $\Delta_r(\mathbf{z}) > \Delta_r(\mathbf{s})$, then $\Delta_r(\mathbf{z}) - \Delta_r(\mathbf{s}) + \gamma \mathbf{E_{s}}_{r+1} \geq 1/2$. In either scenario, with probability at least $1 - \exp(-N)$ next character is ')'. This concludes the proof. $\qquad\square$

For completeness, and for presenting a model with the fewest parameters, we also construct a model as in Eq. (1) with no positional embedding, capable of completing a $\mathcal{D}_{2N}$ prefix into a valid $\mathcal{D}_{2N}$ word. This model is not able to generate different words, it is limited to generating repetitive sequences. It is analogous to those discussed in Theorem 4.1 with $\mathbf{s} = $ '()()()() $\cdots$'.

**Claim 5.3.** *A embedding* $\mathbf{E} = \begin{bmatrix} 1 - 1/(2N+1) \\ -1 \end{bmatrix}$ *with no positional embedding can form a 1 model that completes any prefix into a word of* $\mathcal{D}_{2N}$ *with probability at least* $1 - 2N \exp(-N)$.

*Proof of Claim 5.3.* With a uniform attention $\mathbf{Q} = 0$, if the partial stack is empty, $\Delta_r = 0$, then $\sum_{i=1}^{r} \mathbf{E_{s}}_i < -1/(2N+1)$ and therefore, thanks to Lemma 5.2 for $v < -2N^2(2N+1)$, with probability at least $1 - \exp(v/[r\{2N+1\}]) \leq 1 - \exp(-N)$, next token is a valid open parenthesis. If the partial stack is non-empty, $\Delta_r > 0$, then $\sum_{i=1}^{r} \mathbf{E_{s}}_i > 1/(2N+1)$ and with probability at least $1 - \exp(-N)$ the next token is a closed parenthesis. Union bound allows to conclude. $\qquad\square$

### 5.3.2 COMPLETIONS OF A DEEP CONTEXT

| $q$ | Training samples | Completion of '((((((((((((((((' | Frequency (out of 100) | Is balanced? |
|---|---|---|---|---|
| 8 | $10^7$ | (((((((((((((())))))))))))))) | 52 | Yes |
| 8 | $10^7$ | (((((((((((((())))))))()))))) | 15 | No |
| 8 | $10^7$ | (((((((((((((())))))))()))))) | 12 | No |
| 8 | $10^7$ | (((((((((((((()))))))()))))))) | 8 | No |
| 8 | $10^7$ | (((((((((((((())))))))))))()) | 5 | No |
| 8 | $10^7$ | (((((((((((((()))))))))()))))) | 3 | No |
| 8 | $10^7$ | (((((((((((((()))))()))))))))) | 2 | No |
| 8 | $10^7$ | (((((((((((((())))))))))()))))) | 2 | No |
| 8 | $10^7$ | (((((((((((((()))))())))))))))) | 1 | No |
| 4 | 1 | (((((((((((((()))()()))(()())) | 34 | No |
| 4 | 1 | (((((((((((((())))()()))(()())) | 24 | No |
| 4 | 1 | (((((((((((((()))))()(()))(()()) | 18 | No |
| 4 | 1 | (((((((((((((())))))())))()(()()) | 11 | No |
| 4 | 1 | (((((((((((((()))))())(()))(()()) | 8 | No |
| 4 | 1 | (((((((((((((())))))())(())()()) | 2 | No |
| 4 | 1 | (((((((((((((())))))())(()))()()) | 1 | No |
| 4 | 1 | (((((((((((((())))))())(()))()()) | 1 | No |
| 4 | 1 | (((((((((((((()))))())))()(()()) | 1 | No |

Table 1: Completion of the deepest sequence '(((((((((((((((' of $\mathcal{D}_{32}$ by two models over 100 independent trials.

### 5.3.3 SPECTRAL ANALYSIS OF TOP VALUE MATRICES OF TRAINED MODELS FOR PARENTHESES COMPLETION

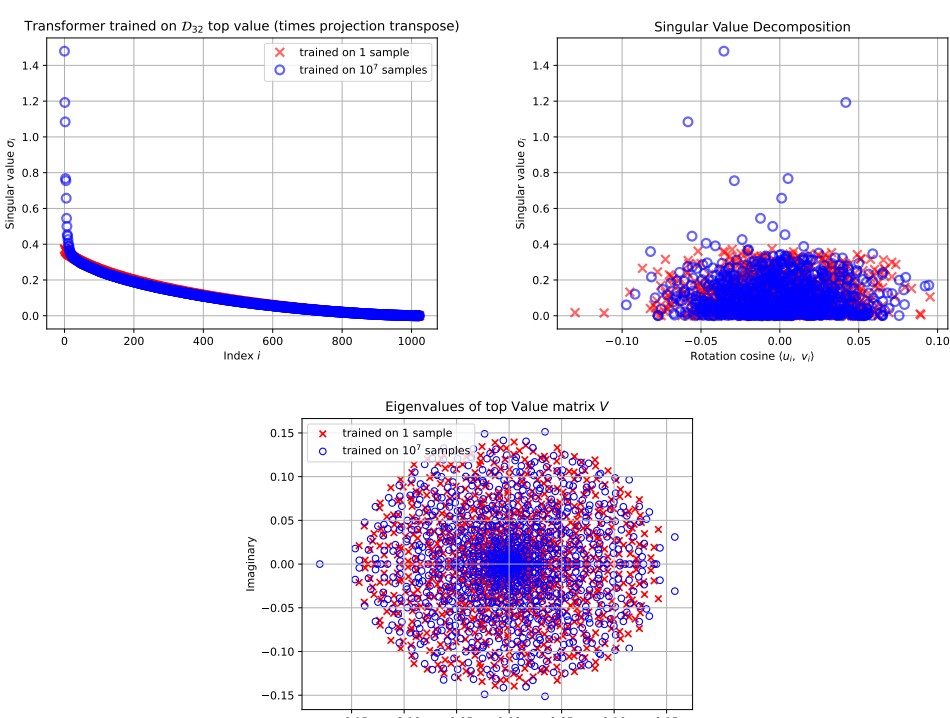

Figure 5: Top value (times projection transpose) matrix for Transformer models trained on 1 and $10^7$ samples. We observe that the model which generalizes exhibits a low-rank structure (top left). Its singular vectors form a negative cosine (top right). The top complex eigenvalue of the model trained on $10^7$ samples pops out with a dominant real negative eigenvalue (bottom).

### 5.3.4 ABLATION STUDY: POSITIONAL EMBEDDINGS AND UNIFORM ATTENTION

**Data.** The set $\mathbf{D}_{32}$ words is here randomly split into a $10\%$ test $(3.5 \cdot 10^6)$ and $90\%$ training set $(\approx 3.2 \cdot 10^7)$. We trained models on sub-samples of $4^k$ for $k = 1, \cdots, 12$ $(4^{12} \approx 1.7 \cdot 10^7)$ distinct words from the training set and compare their performance. We report accuracy of single pass models.

**Models.** We trained 5 models on the same data and compare their performance reported in Figure 6.

1. The largest model $d = 1024, L = h = 4$ of Section 4.2. All parameters of the model are initialized at random and trained using AdamW on the training data (same hyperparameters as in Section 4.2). This model is represented in red in Figure 6.

2. In order to isolate the contribution of positional embeddings, we also train a model with no positional embedding but otherwise identical (represented in black).

3. We fix parameters $\mathbf{Q} = 0$ (uniform attention) and $\mathbf{E} = \begin{bmatrix} 1 & \cdots & 1 \\ -1 & \cdots & -1 \end{bmatrix}$. These models, represented in blue, have att and $\mathbf{E}$ parameters closer to the theoretical study in Section 4. They exhibit strong performance with small training sets, highlighting the benefit of this inductive bias.

4. In order to disentangle the contribution of $\mathbf{B}$ from $\mathbf{V}$ and subsequent linear transformation in the projection and MLP, we train a model where we also freeze $\mathbf{V} = \mathbf{I}_d$ and do not use the MLP. These models are represented with green curves, and mostly under-perform others. We are interested in comparing positional embeddings they learn with a model where $\mathbf{V}$ is also learned (see below).

5. We train a single head model with uniform attention and $\mathbf{E}$. The model uses residuals and layer normalization for efficiency. This model is the closest to our theoretical analysis. We trained it for visualizing learned positional embeddings in Figure 7.

**Observations.** We observe that the fully trained model (1), in red, requires the largest training data, where it is eventually superior to all other models. Haviv et al. (2022) questioned whether a causal transformer trained with no positional embedding can be competitive with a model with positional embeddings. Our controlled dataset study indicates that positional embeddings are pivotal for completion correctness. As shown in Figure 6, comparing red and black full curves suggests that positional embeddings are crucial for achieving higher completion accuracy and lower loss. Model (5) outperforms the highly constrained multi-layer model (4) which has fixed uniform attention at each layer and head. Overall, model performance aligns with the number of learned parameters. An interesting observation is the high performance of model (2) which has fixed uniform attention and $\pm 1$ tokens (in blue). With high embedding dimension, in this two token problem, the value matrix drives performance.

We show in Figure 7 the learned positional embedding's cumulative sums, for models trained with constant uniform attention and interpretable $\pm 1$ token embeddings discussed in Section 4, in dimension $d = 1024$. These positional embeddings were trained on the broadest training set with $4^{12}$ distinct sequences. To enhance visibility, we also run $k$-means on the cumulative sum positional embeddings with $k = 8$ and plot cluster centroids in different colors, and the number of curves assigned to each centroid in the same color. We observe that in the model where $\mathbf{V}$ is learned (blue curves in Figure 6), cumulative sums of positional embeddings do not all end near the origin (top figure). Centroid curves display this distinctly. When $\mathbf{V}$ is fixed to identity, in the middle plot, the transformer's positional embeddings seem to remain closer to the origin, especially at the end of their trajectory. This behavior is even more pronounced in the 1 model (bottom figure), which is the model we studied in Section 4. As a reminder, our theoretical analysis constructed a set of positional embeddings which (opposite curve) follow the token embeddings' cumulative sum, with a $\gamma = -1/2$ value, see Figure 2. Since the cumulative sum of token embeddings is constrained to end at $y = 0$, then the cumulative sum of the constructed positional embeddings end within $1/2$ of the origin. This behavior of the trained model matches our theoretical finding. We posit that the spread of positional embeddings above and below the $x$-axis can be related to layer normalization. We observe that the fully trained models outperform models with fixed token embedding models which outperform models with also fixed $\mathbf{V}$. This suggests that not only positional embeddings is crucial for performance, but the non-trivial interaction between $\mathbf{V}$ and $\mathbf{B}$ when both are learned under no constraints contributes significantly to the model.

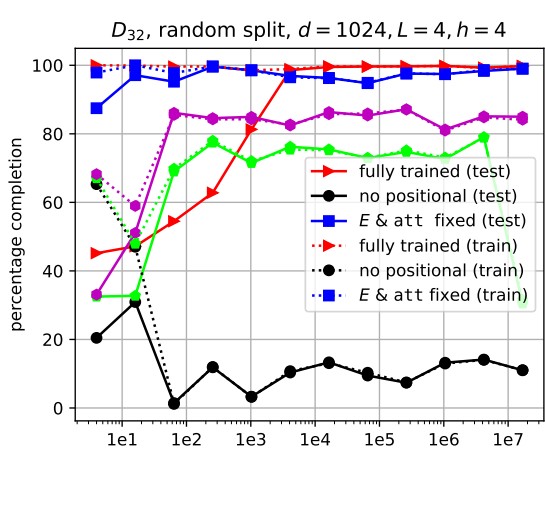

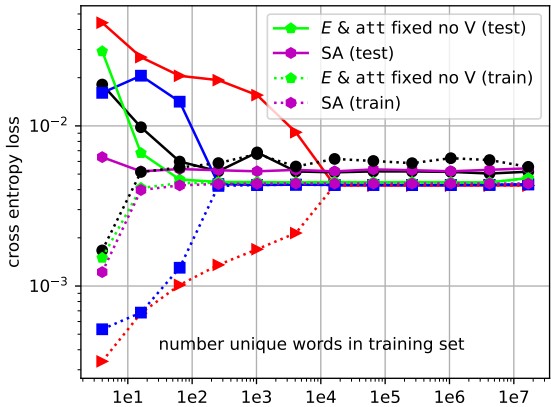

Figure 6: Performance of transformers with (red) or without (black) positional embedding, with uniform attention (blue), and no **V** or MLP after attention (lime) and self-attention only as in Eq. (1 ) in purple trained on $\mathcal{D}_{32}$ datasets with varying number of unique words. Full lines represent out-of-sample performance and dashed line are in-sample.

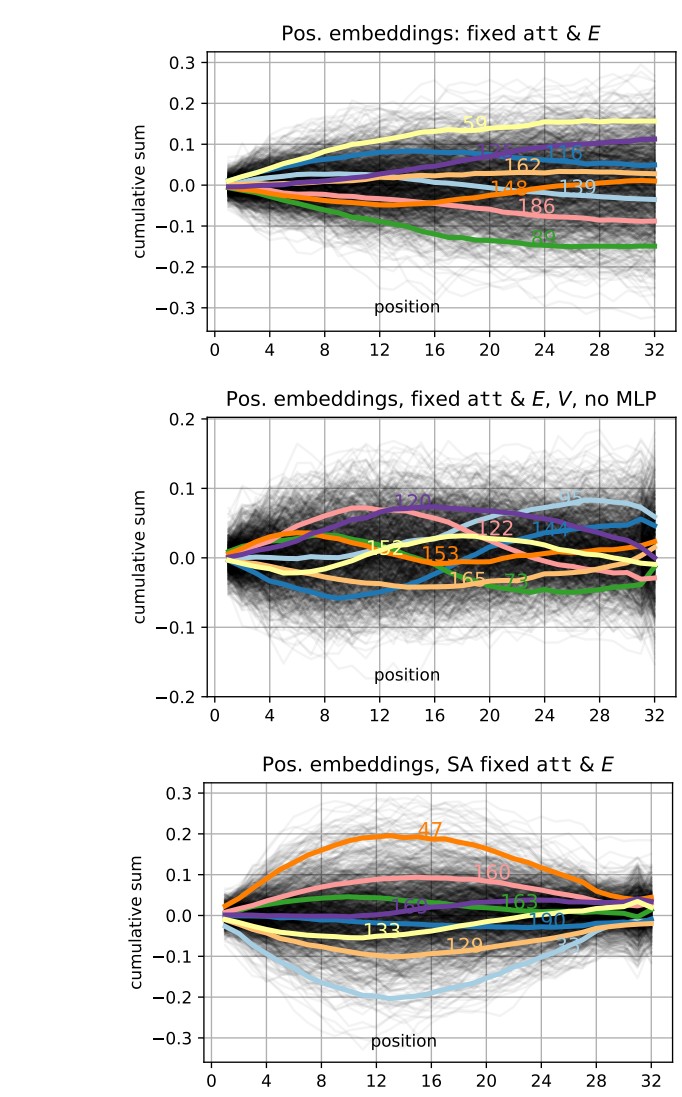

Figure 7: Learned positional embeddings and their $k$-means centroids for $k = 8$. Top: transformer trained with fixed uniform attention and $\pm 1$ token embeddings. Middle: same as above, with $\mathbf{V}$ fixed to identity and no MLP. Bottom: single layer self attention model.

