# OpenReview forum: "Depth Extrapolation of Decoders Trained on Nested Structures"
_ICLR.cc/2025/Conference — Submitted to ICLR 2025_

### Official Review · Reviewer_Ncd5 · 2024-10-25

**Soundness:** 1
**Presentation:** 1
**Contribution:** 2
**Rating:** 3
**Confidence:** 3

**Summary:**

The authors investigate at what level decoder-based transformers handle nested formal statements, with an example on boolean logic simplification. They train a small (8 layers) decoder-based transformer that out-performs larger models, and show it relies on memorization.
The authors develop a theory of memorization in a similar toy case, and use it to develop a closed-form model that, given a single training example generalizes and outperforms larger pretrained models on Dyck language related tasks.

**Strengths:**

The ideas presented in the paper seem interesting.
For example, the authors' framing of memorization into a toy closed form model seems original and interesting.
Another positive example for originality is the usage of a specific problem with varying levels of difficulty (i.e. the simplification of boolean expressions with varying depths) to test large models for generalization is neat. I am personally unfamiliar with other works who did this (but this isn't my specialty).

**Weaknesses:**

The paper is written rather poorly, and is difficult to follow. This is exemplified in several points -

**1.** In the abstract:
    1.1. The main points in the abstract are not clear. Is the main contribution the closed form presentation of memorization? Or the application of this on the nested sequence modeling? I think a better, more ordered presentation is required.
    1.2. Generally, the abstract contains sentences which is unclear how they relate to each other. For example, it is unclear how the sentence "We elicit relations between positional and token embeddings, which explain how large embedding dimensions scale with context size and number of topics in the training set." relates to the previously-presented nested formal statements. Another example is the last two sentences of the abstract, which mean the same thing (at least to my understanding) but are connected with a contradiction ("However"), and it's unclear how they relate to earlier parts.

**2.** Some experiments presented in the paper do not reveal new information, and it is unclear how they relate to other sections. For example, the goal of the analysis in section 2.2 is unclear. I think that the statement by the authors "This finding suggests that training data must contain samples as close as possible to the test set" isn't new, and it's unclear how the results in this section are related to following sections.

**3.** Unclear notations make it difficult to follow the complex claims in the paper - for example, γ appears first in line 232, but undefined anywhere near there. Another example is the matrix V, shown in line 203 (presumably the value weight matrix of the attention mechanism) but not properly defined anywhere. The authors also mention concepts early on (e.g. "a single attention head model") but don't describe them until much later (and sometimes not at all).

**4.** There are many irrelevant references - e.g. in line 104 - "See Appendix" has no valid reference to any appendix.

The above points make it **very difficult to closely follow the lemmas and proofs to verify their correctness**, as well as to understand the main claims made by the authors.

Another weakness is the experimental soundness - many experiments (the ones I was able to follow) don't seem to be rigorous. For example, the experiment of measuring large model accuracy on boolean logic simplifcation (line 104) reports 90% accuracy, but in the appendix the authors mention they only tested 10 samples with depth 1 and only 5 samples of depth 3, which **isn't enough test samples to say anything meaningful**.

In addition, another weakness is the application of these supposed findings on real cases, other than the simple toy case (of a single attention block with a specific type of positional embedding). There doesn't seem to be any implication it will generalize to real models.

**Questions:**

1. In section 2.2 the authors claim that language models often return wrong simplifications of boolean expressions. Did the authors verify permutation equivariance? e.g. a simple case - (p3 & p2) should be considered the same as (p2 & p3). If so, this should be explicitly stated.

2. In section 2.1, why did the authors choose these specific values for the model? Were other settings, that might lead to better performance, considered?

---

> ### Author Response · Authors · 2024-11-22
>
> **W1**. Our revision tries to simplify the message of the paper. We rephrased the abstract to emphasize on the main points (1) Transformers can overfit and memorize but then they fail at extrapolation (2) this is not universal as we prove and numerically verify with our closed-form model (3) but gradient-based optimization seem to be biased against finding those good solutions.
>
> **W2**. The revision tries to better articulate connections between sections. Section 2 establishes that Transformers can memorize / overfit. Then we ask "is this universal". Using Section 3 (theory) we prove in Section 4 (Dyck) that a closed-form model can extrapolate. But experiments of Section 4 show that a gradient-trained Transformer on this problem fails at extrapolation. To our knowledge the problem of depth extrapolation has not been studied rigorously. Our results and experiments provide new perspectives on this problem which is at the heart of reasoning capabilities.
>
> **W3**. Notations are all defined in section 3.1. V is defined in line 195 of the old version and line 184 of the new version. It is the value matrix of size $d \times d$. $\gamma \in \mathbb R$ is a real valued parameter.
>
> **W4**. We added section numbers to the Appendix and references to the Appendix have section number pointers in the revision.
>
> **W5**. LLMs performance on these tasks is simply anecdotal because we don't know on which data the models have been trained. We discuss this in the intro where we state that very often evaluation of LLMs ends up being on non-independent data. The motivation of our work is to report performance when we control train-test split based on depth. This allows to examine extrapolation properly.
>
> **Q1**. LLMs performance evaluation is out of scope for reasons mentioned above. We simply reported comparisons anecdotally to highlight that these are difficult tasks. The revision removed numbers to avoid confusion.
> We did not look at permutation equivariance here. This is an interesting topic to study. I believe that the literature on symbolic reasoning is related to this topic, which is outside the scope of our work.
>
> **Q2**. The revision reports values with larger embedding dimension, but on a larger and harder test set. The exact values do not matter. The message from this section is: Transformers can fit the data. But when they do they overfit and do not extrapolate. In the rest of the paper we question whether this is universal (it's not) and what causes it (possibly gradient-based training).

---

### Official Review · Reviewer_zPw7 · 2024-10-31

**Soundness:** 2
**Presentation:** 1
**Contribution:** 2
**Rating:** 3
**Confidence:** 3

**Summary:**

This papers considers decoder transformers trained on boolean simplification and dyck language completion tasks. Evidence is given that models trained on those tasks often fail to extrapolate to unseen expression depths.
For a dyck language completion, an analytical solution for a 1 layer, 1 head transformer with 1 embedding dimension is derived, capable of arbitrary depth generalization, and its stability properties are analyzed.

**Strengths:**

- I enjoy the analytical transformer solution, showing that this problem should indeed be solvable with the given architecture.
- Figure 1 and the associated analysis on stability given $\lambda$ is insightful.
- The approach to try to minimize the design space is generally pleasing, although that comes with a weakness (bullet 1)
- I like the consistent formatting of showing “Training setup”, then a Figure, and then “Observations”. This is something I will employ in future papers.

**Weaknesses:**

- I question the validity of restricting the general attention mechanism (like setting $Q = 0$), where the only constraint on the restriction is that there still exists a solution in the resulting function space. One has to be very careful what kind of conclusions can be drawn from that. This procedure is baking in (possibly extremely useful) inductive biases that a model has a hard time learning with finite data.
- Related to bullet 1: Something I am missing here are the ablation studies of piece by piece hard-coding the inductive biases used to define the generalizing 1-d model to see which of those biases are important for model training. Those biases are:
1. Setting embeddings to -1 and +1 (or opposite sign vectors of the correct dimension)
2. $Q=0$
3. $V=vI$
- The writing is quite confusing. Section 2 and 3-5 seem somewhat disconnected. I don’t need a boolean simplification example to introduce dyck language completion, it seems like “fluff” to make the paper longer. I was often confused while reading, and only sometimes had “aha” moments much later. The evaluation protocol in Sec 4.2 is very unclear.
- I find the evidence in Fig 3 unconvincing. The second largest singular value is positive. Also it looks like this is only one model. For different seeds, is the largest singular value always negative?
- After having read the paper, I am unsure what to conlude. What knowledge am I supposed take away from this paper? On a high level, it seems “models fail to extrapolate in some way (in this case nesting depth), even though they have computational capability to succeed”, which I don’t think is a very novel conclusion at this point. However, breaking it down to a setup this minimal is a good step!

**Questions:**

- Table 1: Are these results from training q≤N or only q=N? Your table suggests the latter while the “observations” section suggest the former.
- Table 1: it is quite interesting that train q=2/test q=3 in-sample is (top-left) is worse than train q=5/test q=6 oos. Do you know why?
- You say you train on 7.3e10 tokens in 7.2e7 sequences. Do you somehow make sure the #tokens/ seq is the same for different q? Or do the numbers refer to the union of all training sets? I’m wondering whether my previous question can be explained by an increased number of training tokens
- Fig 2 (left) is on a held out sample? how big is the training set there?
- I am not sure I grasp the evaluation protocol of Fig 2 (right). Training happens on q≤4,8,12, and the evaluation is actually on q ≥ 9 (or 12? or 13?) Please make this clearer, I am very confused.
- Page 8, “Observations”: Except for d=2, it does not look to me like the models’ performance clusters by embedding dimension. Where do you see that?
- It is unclear to me which singular values you are plotting in Fig 3. The “value matrix (times projection transpose)”. What is that? How does a low rank of this thing indicate $V=-vI$ ? I think it helps if you write that down in the section.

### Nits:

- Is it normal that the line numbers are not well aligned with the text? I give approximate line numbers.
- L192:  $E_{s_{i,:}}$  is indexed incorrectly
- it is kind of weird to denote the set of $\mathbf{s_i}$ as $\mathbf{s}$, since that set should also have an index $r$
- the explanation on why $\lambda < 0$ attracts and $\lambda > 0$ repulses could be clearer, you explain it better on Page 15 “Generalization”, but please link to that in the main text (sec 4.1).
- Fig 2 (left) is missing a y axis label. Was the intention to share the label with Fig 2 (right)?

---

> ### Author Response · Authors · 2024-11-22
>
> **W1**. The revision simplifies and emphasizes the bold lines of our reasoning and why we use strong hypotheses on the theoretical results. As a high level summary: we show on the Boolean expression simplification problem that Transformers are able to fit the data. But they seem to overfit and therefore fail at generalizing. Is this behavior universal? to prove that it is not it is sufficient to prove that a simple Transformer can extrapolate in a simple nested expression problem. This is why the model we consider is simplified and this is why in a second set of experiments we work with a simpler problem.
>
> **W2**. Ablation study is added to the Appendix 5.3.4.
>
> **W3**. Thanks for the feedback. We have tried to highlight the reasoning in the revision. The Boolean expression problem demonstrates that the model can fit the data but fails at extrapolating. The simple model shows that Transformers contain solutions that can extrapolate. Numerical experiments suggest that gradient-based solutions do not have a favorable bias for finding those solutions.
>
> **W4**. Singular values are by definition always positive. We added complex eigenvalues of this square but not symmetric matrix as a plot in Figure 5, Appendix 5.3.3. You can see on this Figure that the eigenvalue with the largest modulus of the 10^7 model (in blue) is the one with a real negative value.
>
> **W5**. The revision tries to address your concern on clarifying the take-home message: "there exists Transformers capable of extrapolation, but the current gradient-based methods seem to fail at finding them." We believe that (1) framing the extrapolation problem helps fairly evaluating reasoning (2) characterization of extrapolators is an interesting contribution, which opens questions for future research on optimization
>
> **Q1**. $q\leq N$. The revision simplifies the presentation and more clearly call out the take-home message.
>
> **Q2**. Fluctuations were probably due to small test sets. We used a 10x larger test set in the revision. Fluctuations of the out-of-sample q=6 model around 1% accuracy are not really concerning here: the message is that we can reach ~ 70% accuracy on this difficult task, but when we overfit. On out-of-sample extrapolation results are poor. That's what is important to retain from this section.
>
> **Q3**. The mini-batches are sequences starting from <start> and of fixed length. Therefore all models are given the same number of input tokens which may correspond to a different number of expressions. It is an interesting remark you are making that if the problem is viewed as a supervised learning problem (with y being the RHS simple expression) then the number of equalities is lower for deeper expressions. More broadly, the problem of collusion of depth and length generalization is one that we did not address in this work.
>
> **Q4**. Training set has 2.4e7 sequences, see l. 370 of the revision.
>
> **Q5**. Training is on $q \leq 4,8,12$. Evaluation is on $q \geq 13$. See l. 396 in revision. We corrected a typo (12--> 13; thank you) and clarified the setup.
>
> **Q6**. Performance grows with embedding. Corrected in revision l. 385.
>
> **Q7**. Transformer models multiply value by the transpose of another matrix called projection. From a model standpoint $V P^T$ is what we are studying when we look at the value matrix. However, when you optimize the model with backprop, it is advantageous to have the two separate linear models. When we study the spectral properties, those are captured in $V P^T$, not in $V$.
>
> Nits.
>
> **N1**. NA
>
> **N2**. Indexing is correct. We are referring to the row index $i$ of $E_s$ which is by definition the same as row $s_i$ of $E$.
>
> **N3**. $s$ is a sequence not a set. We are using $r$ when we refer to the length of a context. $n$ is the maximum context window. $i$ is a moving index between 1 and $n$.
>
> **N4**. We elaborated on the role of $\gamma$ in the revision l. 240 "Stability"
>
> **N5**. y axis label is not missing. It's accuracy and shared with the plot on the right. ticks are on the right as we have the legend on the left.

---

> > ### Comment · Reviewer_zPw7 · 2024-11-26
> >
> > Thank you for your rebuttal.
> >
> > For future rebuttals, could you mark your changes in a color, otherwise I have a hard time to follow what I originally was concerned about. I could not find where specifically you changed things to address W5 and W3, esp since I don't seem to be able to access the old version of the paper anymore.
> >
> > Your reasoning in W1 makes sense to me.
> >
> > In Fig 1, IIUC, you train a model with up to depth q (defined by x axis) and then test for IS and OOS accuracy. How is there q=6 in sample data for a model trained on q=3? Did i understand wrong that in-sample means part of the training data?
> >
> > Thank you for the ablation study.
> > Is the caption correct in the figure 6? It quotes lime twice and misses purple.
> > I wonder, did you run a model where Q=0 but not E=[-1, 1]?
> > Do you know why blue is eventually outperformed by red? presumably these are good inductive biases. When do they fail?
> > What happened to the green curve for 1e7 data?
> > How do you interpret the failure of the model without positional encoding? It seems to align with your theory (the cumsum implemented by positional encoding), but it also seems that the models other than Figure 7 (bottom) don't actually seem to follow the cumsum. Also, IIRC models with only the causal mask as positional bias are able to count as well.
> > Is model (5) in the plot?
> >
> > Re W4: Sorry i didn't mean negative singular value, but singular value that lies on a negative cosine (x axis fig 5) I was interested in whether this plot holds up (largest SV in the negative cosine regime) for different seeds (same for Fig 5 bottom).
> >
> > Thank you also for answering my questions. I don't think i fully understand the answer to Q3. In a minibatch, they are padded to the same length? Or do you sample them such that you have only the same length? Can you point me to where you describe this in the paper?
> >
> >
> > The main and overarching limitation that I have remains. The takeaway from this paper to me is that models cannot generalize to something that extrapolates in some way, i.e. some distribution shift. I don't expect a model to be able to do that. Maybe other reviewers care to comment on this, if they feel strongly positive about this paper.

---

> > > ### Author Response · Authors · 2024-11-26
> > >
> > > > The takeaway from this paper to me is that models cannot generalize to something that extrapolates in some way, i.e. some distribution shift. I don't expect a model to be able to do that.
> > >
> > > The takeaway is more nuanced than "models cannot extrapolate". The model class contains local minima which can extrapolate but the class of commonly used gradient-trained models initialized at random are not providing those desirable solutions.
> > >
> > > We also expected models to fail at reasoning on harder logic than training set. But then promises of AGI and papers such as "sparks of intelligence" caught our attention. When we read those statements, we could not clearly see if test sets were independent from training sets. This is not systematically because authors have an interest in making those claims. It is also because it is hard to know what exactly is in the very large training sets.
> > >
> > > For this reason, we propose to study a very simple instance of reasoning where we isolate one complexity measure of reasoning difficulty: depth. Our goal is to confront our initial intuition (aligned with yours) against results from well-curated test sets and theory that confirms or rejects our prior intuitions. It turns out that the situation is more complex than we expected.
> > >
> > > > Fig 1 IS / OOS definition
> > >
> > > line 126  "We define as out-of-sample all expressions with simplified form (right-hand-side of equality) in a held-out set of 128 randomly selected expressions "
> > >
> > > **Ablation**
> > > We added loss function values to complete the picture.
> > >
> > > > Ablation study colors in caption
> > >
> > >  second one was purple. Thank you.
> > >
> > > > What happened to the green curve for 1e7 data?
> > >
> > > Loss function did not decay for as long as the other models, it returned nan and training stopped. Since the model was not top performing we did not re-start training and simply shared the values obtained before numerical errors propagated.
> > >
> > > > Do you know why blue is eventually outperformed by red? presumably these are good inductive biases. When do they fail?
> > >
> > > red has more free parameters. It has higher sample complexity but also higher capacity. When large enough data is fed to the model it eventually gets better than a (well) constrained model
> > >
> > > > IIRC models with only the causal mask as positional bias are able to count as well.
> > >
> > > They can count but they cannot complete parentheses because they need more than counting the stack for that: they also need to know how far they are in the string. If not they may keep adding open parentheses (stack is still positive). What we observe with that model is that parentheses stack are often open at the end of the 2N = 32 characters.
> > >
> > > In Claim 5.3 in appendix we build a single head attention model with no positional embedding which correctly completes parentheses. This model is hardly constrained to have the lowest open parentheses stack possible at all times. It is the only non-PE model we know which accomplishes this task. Is it equivalent to the PE model with our closed form for B derived from the sequence `()()()()...`
> > >
> > > >  Is model (5) in the plot?
> > >
> > > That's the purple SA (for Self-Attention) model
> > >
> > > **W4** We did not cherry pick the data shared here, but also did not re-train for multiple seeds. Similar findings have been reported by other authors as referred in the paper.
> > >
> > > **Q3**  Sequences have different lengths. We align the minibatches to start at <start> and be 1024 tokens long. The minibatches do not necessarily end on a <end> token. See l. 130 of the last version.

---

### Official Review · Reviewer_nqML · 2024-11-01

**Soundness:** 3
**Presentation:** 3
**Contribution:** 3
**Rating:** 6
**Confidence:** 3

**Summary:**

This paper studies the ability of transformers to extrapolate reasoning on harder tasks by studying their behaviour on boolean expression and dyck languages. The authors first show that transformers are can not reason beyond the given depth q of a clause, that is, while the transformer models do quite well on "in-sample" data, i.e, data that matches the same complexity (in terms of depth and number of clauses) as the training data, the fails to extrapolate the reasoning beyond that.

Next, the authors study the ability of transformer models to study Dyck languages with a single layer and single head Transformer model. Here the point is to show the effects a learned position embedding could have towards memorization of LLM towards the input data.  The authors show that when training the models over one data distribution, under the conditions when Q = 0, based on the value, the learned position embeddings can sometimes force the model to complete a sequence in a way that it sees during training.

**Strengths:**

- The observation on training over more difficult depth enables better performance over relatively simpler depth $q$ is quite interesting!

- The result in Theorem 3.1, which shows that for appropriately scaled value $v$ and $\gamma$ parameter, the model can either correctly predict the sequence, or get worse (i.e, in the case when $\gamma$ and $v$ have opposing signs and are small).
- The experiments with dyck languages and how the position encoding effects the training of the model is very interesting. The authors show that sometimes the learned position encodings could be the reason why the model chooses to finish a clause the way it does. That is, it memorizes.

- Through their experiments on deeper layers and larger embedding dimensions the authors show that they learned models when trained on sufficiently large number of samples are able to do well on validation set.
- The observation on the fact that the value functions can be approximated by $-vI$ is pretty interesting. esp given the theory results.

**Weaknesses:**

- I am curious as to why we expect a model like GPT to reason about and do well on the reasoning for boolean clauses task, especially when the depth increases. That is not what the model is trained on, and could also be quite a difficult task for humans to solve. I realize that this is not the main focus of the paper, but expecting an LLM to do well on a boolean logic simplification task seems to be asking for a bit much.
- It seems like the theoretical setup is too simplistic.
	- The authors make a key assumption here that the Query is 0. If the Query is not zero, would the dependence on the position encodings still be as great, esp if the inner product between the query and value is high?
	- The results on multiple sequences assume that the tokens in each sequence are also independent, which basically is a small extension of the single sequence result.
       - How would tokens that are not iid change the results?
- What is the intuition of $\gamma$? what effects does it have on the position encodings? Can the authors elaborate on that?
- Some of the experimental results are not surprising (given their lack of ability to also length generalize), i.e, the generalization to deeper depth getting more difficult as the training data complexity is further and further away from the testing data complexity.

**Questions:**

- some of the notation and terms used is a bit unclear. Where does $\gamma$ come from? Is $\gamma$ something that the authors *fix* themselves, i.e, is part of the constructed solution to the position encodings?
- The authors use the term out-of-sample basically validation set (i.e, data with the same properties as the input data) and in-sample samples from the training data?

---

> ### Author Response · Authors · 2024-11-22
>
> **W1**. Indeed this is a difficult task for humans and machine alike. This is the first sentence of the abstract. Since the scientific community is motivated to use AI for solving problems that humans fail at solving, it belongs to ML researchers to (a) measure when and how such claims are actually valid (b) build models that can actually solve such problems. We propose a methodology to examine the difficulty of reasoning in nested structures, if the problem difficulty goes beyond the difficulty of already solved problems.
>
> **W2. a)** The goal of our theory is to prove that the finding "Transformers can solve nested problems when they overfit" is not universal: there exists Transformers and nested problems where we observe extrapolation. For this, it is sufficient to prove that a simplified model extrapolates on a simple problem. We also show limitations of current methods at identifying such solutions. Our work opens research directions to find models capable of extrapolation. This can be achieved either using better optimization or initialization methods or using better inference routines.
>
> **W2. b)** Multi-sequence in all generality is a difficult problem. To avoid misleading the readers attention, in the revised version of the paper we discuss the problem at a high level, provide a reference and postpone that study to future work, see lines 250, 473.
>
> **W3**. We added a discussion on $\gamma$ in the revised paper, please refer to line 240 on *stability*: $\gamma, v < 0$ result in a stable model. The resulting Dyck completion model can extrapolate. In contrast, $\gamma, v>0$ builds a model capable of interpolating (memorizing) which fails at generalizing.
>
> **W4**. It was surprising to us that the gradient-optimized model on a single sequence did not have the same extrapolation properties as the closed-form model. The train-test split along problem complexity is a methodological contribution which can help in other reasoning learning problems.
>
> **Q1**. $\gamma \in \mathbb R$ is a free parameter. The revision contains more details on how its value impacts the model's behavior.
>
> **Q2**. For Boolean expression completion, "out-of-sample" are data with simplified forms in a small held-out set of expressions. None of the data used for training the model had a simplified form in this set.  In-sample are Boolean expressions with right-hand-side expressions within the set of expressions we used for training. Please refer to line 126 of the revision.

---

### Official Review · Reviewer_XghT · 2024-11-04

**Soundness:** 3
**Presentation:** 3
**Contribution:** 3
**Rating:** 6
**Confidence:** 3

**Summary:**

This paper studies how Transformer learns bounded hierarchical language and how the model extrapolates over different depths. They present a theoretical construction of a 1-layer Transfomer with learned position encoding that can memorize a long training data and generalize over any depths. They further show through experiments that in practice, the model learns very different solutions.

**Strengths:**

1. The paper presents an interesting construction for Transformer that can generate the Dyck language, which is very different from any previous works.

2. The paper presents a vast range of empirical studies to justify the theoretical assumptions and also strongly show that the learned architecture is different from the generalizable architecture the paper constructed.

**Weaknesses:**

1. While the construction that the authors presented can successful generate valid dyck sequence for any valid dyck prefix, it will not minimize the pretraining loss. For example, consider a training set that contains multiple sequences (which is the setting in the experiments) and two sequences have shared prefix and different conclusions, the current construction would not be able to minimize training loss. This seems to explain the discrepancy between theory and experiments, and shows that the construction is restricted to the single sequence authors considered.

2. The depth generalization setup is studied in previous works [1] and the authors haven't provided a thorough discussion on how the current results differ from them.

[1] Grokking of Hierarchical Structure in Vanilla Transformers, arxiv.org/abs/2305.18741

**Questions:**

1. How to accomodate the current theoretical result to setting with multiple sequences (see weakness 1)?

---

> ### Author Response · Authors · 2024-11-22
>
> **W1 - Q1**. The solution we provide does minimize the loss function, when the loss is calculated over one sample. This is proven in the main statement of our Theorem 4.1. We have also observed this in our numerical experiments. The loss function with a single solution does not have a single minimizer. Our claim is: the stable solution $\gamma = -1/2$ is capable of extrapolation. The other solutions $\gamma >0 $ do not extrapolate. The solutions identified by gradient descent methods are also incapable of extrapolation.
> You are right that the single sequence solution does not minimize the loss calculated over multiple sequences. Characterizing the minimizer and generalization power of the multi-sequence problem is difficult. We share more on this topic in line 250 of the revised paper.
>
> **W2**. Thanks for bringing this paper to our attention. This paper does not split train-test based on expression depth. Our focus is on depth generalization which they do not study. We added a reference and explained this in line 376 of the revision.

---

> > ### Comment · Reviewer_XghT · 2024-11-24
> >
> > I have read the response. I believe that while characterizing the minimizer of the multi-sequence problem is certainly harder, it is more relevant to the empirical setting. I will keep my score.

---

> ### Author Response · Authors · 2024-11-24
>
> Yes, multi sequence is more similar to the empirical setting. We characterize the behavior of those model with empirical findings, including demonstrating their weakness at depth generalization.
>
> The point of this paper is to study depth generalization. While empirical findings show that multi sequence models do not excel at that, our **theory shows that there exist** Transformer models which minimize a next token prediction loss function that do succeed at depth extrapolation.
>
> The purpose of our theory is **not to explain** what is empirically observed. But our theory **constructs** a Transformer which minimizes a next token prediction loss and also it is an extrapolator. Please read lines 255 - 265 of the paper where we provide sufficient background to the challenges of multi-sequence analysis.
>
> What we learn from the theory here is that there exist more favorable local minima of the loss function that gradient descent is missing.

---

> > ### Author Response · Authors · 2024-11-28
> >
> > Reviewer XghT,
> > Please note the gap between single sequence optimal model discuss in the theory section and the characterization of the local optimum gradient-based methods found. The closed-form model extrapolates (if $\gamma<0$) while the gradient optimized model memorizes and repeats the memorized sequence: compare Figure 2 for closed-form and Figure 4 for the empirical.
> > In view of this gap, we do not expect the multi-sequence model obtained using the same theoretical tools (first order criterion + JL) to explain more of the empirically obtained model. This is again a case where the theory model proves a counter example and a simple counter example is sufficient to establish the statement that gradient-optimized empirical models are sub-optimal. In the last version of the submission l. 255-265 we elaborate on the multi-sequence situation. It must be clear from reading this that strong assumptions are required to approach the multi-sequence problem. We defer that to future work.

---

> > > ### Comment · Reviewer_XghT · 2024-11-28
> > >
> > > I agree that the current theory shows that there exists a weight configuration that can (1) minimize training loss on a single sequence, and (2) depth generalizes. I also agree that the empirical results show that the weight learned from minimizing training loss on multiple sequences does not depth generalize. My concern is that:
> > >
> > > 1. The current theory does not show there exists a weight configuration that can minimize training loss on multiple sequences and depth generalize.
> > >
> > > a. Therefore the current result could not justify the statement that gradient descent misses some good local minima.
> > >
> > > b. Also, because a minimizer that can solve Dyck language with depth $q$ automatically solves Dyck language with depth less than $q$, this result seems equivalent to saying that there exists a weight configuration that can minimize the training loss for the Dyck language, which has been shown previously.
> > >
> > > 2. Given 1.b, I think the non-trivial aspect of the current theory is the characterization of how $\gamma$ impacts the extrapolation. However, I could not see the analogy of this when considering training on multiple sequences.

---

> > > > ### Author Response · Authors · 2024-11-28
> > > >
> > > > > The current theory does not show there exists a weight configuration that can minimize training loss on multiple sequences and depth generalize.
> > > >
> > > > It is pointed out in the paper l. 255-265, there does exist solutions for the problem where $E_s+B$ is orthogonal for all sequences $s$ in the training set, where the multi-sequence model memorizes. Studying extrapolation for these is an interesting question. We will need to add more assumptions and state the result clearly. It won't fit in this paper which already has a complete loop of argument that does not seem to be straightforward to grasp.
> > > >
> > > > >  the current result could not justify the statement that gradient descent misses some good local minima.
> > > >
> > > > the current result **does justify** that gradient descent misses some good local minima. In order to prove an existence result it is sufficient to build one example. In this paper, we prove with theory and numerically that the favorable local minimum has the extrapolation property. We show the performance and we display examples of sequences generated by the empirical model trained on a single sequence which does not exhibit that nice property. This is a sufficient proof. Multi-sequence will be nice to have but the proof is already established with a minimum setup.

---

> > > > > ### Comment · Reviewer_XghT · 2024-11-28
> > > > >
> > > > > I thank the author for the clarification. I think it would improve the paper to better clarify the motivation for studying loss on a single sequence in future versions. I have raised my score to a positive rating.

---

> > > > > > ### Author Response · Authors · 2024-11-28
> > > > > >
> > > > > > thanks for all your feedback and suggestion which we incorporated in l.  210 "...  it is sufficient to prove that a simplified model fulfills the property. Hence *we will restrict our study to a single sequence loss, which will provide an extrapolator as demonstrated in Theorem 4.1*"

---

### Author Response · Authors · 2024-11-22
**Rebuttal and submission revision**

We thank reviewers for their valuable feedback. In the revision we (1) more clearly outlined the reasoning and message of the paper (2) simplified the presentation of section 2 and presented results on a larger and more difficult test set which more concisely highlights this section's message (3) we organized the Appendix to address questions raised by the reviewers.

We address individual questions below. **W1, W2...** refer to concerns raised in *Weaknesses* sections and **Q1, Q2, ...** address questions in the order they are asked in the *Questions* section.

---

### Meta-Review · Area_Chair_pzxt · 2024-12-23

**Metareview:**

This paper studies performance of Transformers on boolean simplification and dyck language completion tasks. The paper presents failure of gradient-based training to finding solutions with depth extrapolation. The paper also conducted empirical analysis to verify some of the claims.

The reviews for the paper were on the borderline to slightly negative. The primary concerns of the reviewers were about: (1) assumptions, (2) unclear presentation and (3) weak empirical analysis. Upon closely looking at the paper, I think the results of the paper are certainly interesting. Results in similar spirit was also recently demonstrated for in-context linear regression so overall this line of research is important to investigate. However, I do agree with the reviewers about the assumptions and the unclear presentation. I encourage the authors to address the reviewer's concerns. I recommend rejection in the current form.

**Additional Comments On Reviewer Discussion:**

The rebuttal convinced one of the reviewers to increase the score. However, it failed to concern some existing concerns about assumptions and presentation

---

### Decision · Program_Chairs · 2025-01-22

Reject